# Textual Bayes: Quantifying Prompt Uncertainty in LLM-Based Systems

**Brendan Leigh Ross**\* **Noël Vouitsis**\* **Atiyeh Ashari Ghomi**
**Rasa Hosseinzadeh** **Ji Xin** **Zhaoyan Liu** **Yi Sui** **Shiyi Hou**
**Kin Kwan Leung** **Gabriel Loaiza-Ganem** **Jesse C. Cresswell**
```
{brendan, noel, atiyeh, rasa, zhaoyan,
amy, gloria, kk, gabriel, jesse}@layer6.ai
ji.xin@uwaterloo.ca
```
Layer 6 AI, Toronto, Canada

## Abstract

Although large language models (LLMs) are becoming increasingly capable of solving challenging real-world tasks, accurately quantifying their uncertainty remains a critical open problem—one that limits their applicability in high-stakes domains. This challenge is further compounded by the closed-source, black-box nature of many state-of-the-art LLMs. Moreover, LLM-based systems can be highly sensitive to the prompts that bind them together, which often require significant manual tuning (i.e., prompt engineering). In this work, we address these challenges by viewing LLM-based systems through a Bayesian lens. We interpret prompts as textual parameters in a statistical model, allowing us to use a small training dataset to perform Bayesian inference over these prompts. This novel perspective enables principled uncertainty quantification over both the model's textual parameters and its downstream predictions, while also incorporating prior beliefs about these parameters expressed in free-form text. To perform Bayesian inference—a difficult problem even for well-studied data modalities—we introduce Metropolis-Hastings through LLM Proposals (MHLP), a novel Markov chain Monte Carlo (MCMC) algorithm that combines prompt optimization techniques with standard MCMC methods. MHLP is a turnkey modification to existing LLM pipelines, including those that rely exclusively on closed-source models. Empirically, we demonstrate that our method yields improvements in both predictive accuracy and uncertainty quantification (UQ) on a range of LLM benchmarks and UQ tasks. More broadly, our work demonstrates a viable path for incorporating methods from the rich Bayesian literature into the era of LLMs, paving the way for more reliable and calibrated LLM-based systems.

## 1 Introduction

Large language models (LLMs) have become increasingly embedded in our daily lives, with growing adoption across domains such as customer support (Chaturvedi & Verma, 2023), code generation (Wang et al., 2021; Chen et al., 2021), scientific research (Boiko et al., 2023; Schmidgall et al., 2025; Yamada et al., 2025), and creative writing (Gómez-Rodríguez & Williams, 2023). As their capabilities continue to advance, there is also mounting interest in deploying them in agentic systems, wherein they perform tasks autonomously on behalf of users (Wooldridge & Jennings, 1995; Xi et al., 2025). Despite their proliferation, however, trust in LLMs remains limited, largely due to their propensity to generate hallucinated content (Maynez et al., 2020; Xu et al., 2024) and their susceptibility to adversarial attacks and jailbreaking (Wei et al., 2023; Zou et al., 2023; Yan et al., 2024). These vulnerabilities in LLM-based systems therefore must be addressed, especially to fully unlock high-stakes domains such as finance and medicine. A key step towards mitigating these risks is to reliably quantify the uncertainty of LLM-based systems. Accurate measures of uncertainty ensure that, when unable to answer, LLM-based systems can abstain, defer to human experts, or augment their context with subroutines based on retrieval or reasoning (Lewis et al., 2020; Wei et al., 2022).

---

\*Equal contribution.

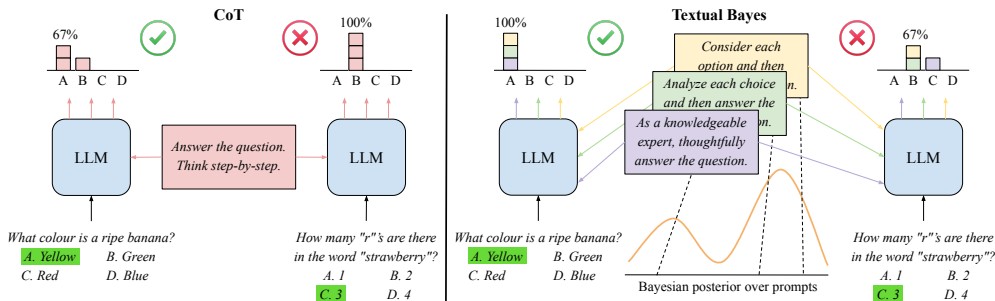

Figure 1: In chain-of-thought (CoT) prompting (left), answers are generated by an LLM using a single fixed prompt; this frequentist approach does not account for uncertainty about how the model should be prompted, causing potential issues such as overconfidence on incorrect answers. In Textual Bayes (right), we sample prompts from our Bayesian posterior and use each to generate answers from the LLM; this allows for principled uncertainty quantification over both the prompts themselves and the resulting generated answers.

Despite recent progress, uncertainty quantification (UQ) for LLMs is far from solved and no consensus exists over exactly what should be quantified (Kuhn et al., 2023; Wang & Holmes, 2024; Yang et al., 2024b). In this work, we propose to better quantify uncertainty in LLM-based systems by viewing them through a Bayesian lens. In light of a model, observational data, and one's prior beliefs, Bayesian inference uses Bayes' rule to compute the distribution of possible model parameters. Commonly applied in classical statistics and deep learning, Bayesian inference is a principled and mathematically grounded approach to UQ (Bernardo & Smith, 2009). Bayesian techniques have led to high-profile successes in methods like variational autoencoders (Kingma & Welling, 2014) and Bayesian neural networks (Blundell et al., 2015). As with their 20th century forebears (e.g., (Duane et al., 1987; Saul et al., 1996)), these methods estimate uncertainty over high-dimensional continuous variables. Here, we bring Bayesian methods into the age of LLMs. In LLM-based systems, the main variables of interest are prompts, since LLMs themselves are often black boxes that can only be accessed via an API. By treating prompts as textual parameters in a statistical model, as illustrated in Fig. 1, we can use Bayesian inference to estimate distributions over their values. These distributions rigorously quantify our uncertainty about the models themselves. Furthermore, they can be integrated into uncertainty estimates on the system's downstream outputs via easy-to-compute Monte Carlo estimates. To the best of our knowledge, we are the first to perform Bayesian inference over the space of free-form prompts in LLM-based systems.

Adapting Bayesian methods to text has its challenges and advantages. On the one hand, textual variables are discrete, making it difficult to apply traditional Bayesian deep learning techniques such as gradient-based Markov chain Monte Carlo (MCMC) (Welling & Teh, 2011) or variational inference (Saul et al., 1996). We address this obstacle with a novel text-based MCMC method: *Metropolis-Hastings through LLM Proposals* (MHLP). On the other hand, textual variables are better suited conceptually to Bayesian modelling than high-dimensional continuous variables such as the weights of a deep neural network. Bayesian inference famously requires the specification of prior beliefs about a variable; textual variables are more amenable to human priors than neural network weights, and as we show, prior beliefs can be readily incorporated into LLM-based systems as free-form text.

To advance and justify our proposed method, this work contains the following contributions:

1. We take a novel perspective on LLM-based systems in which prompts are viewed as Bayesian textual parameters $\theta$ in a model $p(y \mid x, \theta)$. We show how this formulation leads to a principled way to incorporate prior beliefs about $\theta$ while quantifying our inherent uncertainty in the model.

2. To implement our Bayesian approach, we propose *Metropolis-Hastings through LLM Proposals* (MHLP), an MCMC algorithm to sample from intractable distributions over textual variables. MHLP has broad potential applications even beyond Bayesian inference.

3. We propose a novel metric of model calibration, *semantic expected calibration error*, for quantifying calibration, a form of UQ, on free-form textual outputs.

4. We systematically evaluate our method through standard LLM benchmarks and baselines, showing that it improves performance while providing state-of-the-art UQ over model outputs.

## 2 BACKGROUND AND TERMINOLOGY

### 2.1 LLM-BASED SYSTEMS

The central object in this work is the LLM-based system. The most common LLM-based system is one consisting of a single input $x$ (e.g., a question), a prompt $\theta$ (e.g., a system message defining instructions for the model's behaviour, shared across all $x$), and an output $y$ (e.g., the model's predicted answer), which we denote

$$y = \textbf{LLM}(x; \theta). \tag{1}$$

We allow $\textbf{LLM}(x; \theta)$ to be any open- or closed-source model that we view as a *random* function of $x$ and $\theta$, whose randomness depends on the underlying LLM sampling strategy (e.g., greedy, temperature, nucleus). In general throughout the work, capitalized, boldface function names will indicate random functions comprising one or more LLM calls.

LLM-based systems can be more complex than single-prompt models. Many recent works have proposed to group LLM calls of arbitrary count and complexity into pipelines parameterized by the prompts used at each step (Khattab et al., 2024; Zhuge et al., 2024; Yuksekgonul et al., 2025; Cheng et al., 2024; Hu et al., 2024). For example, Self-Refine (Madaan et al., 2023) iterates on an initial LLM output by alternating between an LLM call providing feedback and one incorporating the feedback into refinement. Fully agentic systems integrate multiple LLM and tool calls to arrive at a final output. In full generality, we can describe a forward pass through an LLM-based system as

$$y = \textbf{LBS}(x; \theta), \tag{2}$$

where $\textbf{LBS}(\cdot; \theta)$ can be described as a directed acyclic graph with $k$ edges in which each edge $e_j$ corresponds to an LLM call $\textbf{LLM}(\cdot; \theta_j)$ parameterized by a prompt $\theta_j$ and where we denote the combination of all prompts in the system as $\theta = (\theta_1, \ldots, \theta_k)$. Since each LLM call in the system is potentially random, $y$ is a random function of $x$ parameterized by $\theta = (\theta_1, \ldots, \theta_k)$. The LLM-based system thus forms a statistical model for $y$ whose probability mass function we express as $p(y \mid x, \theta)$, where sampling $y \sim p(y \mid x, \theta)$ is equivalent to computing $y = \textbf{LBS}(x; \theta)$.

Unlike a linear regressor or neural network where $\theta$ denotes continuous model parameters, for an LLM-based system $\theta$ denotes *textual* parameters. From the statistical modelling perspective, a natural next step is to find the optimal value of $\theta$. For example, given an i.i.d. dataset $\mathcal{D} = \{(x_1, y_1), \ldots, (x_n, y_n)\}$, one might want to perform maximum-likelihood:

$$\theta^* = \arg\max_\theta p(\mathcal{D} \mid \theta) = \arg\max_\theta \prod_{i=1}^n p(y_i \mid x_i, \theta). \tag{3}$$

The discrete nature of textual parameters prevents us from applying gradient-based algorithms to maximize likelihood in LLM-based systems. Prompt engineering can be understood as approximating $\theta^*$ by having a human propose candidates $\theta$ until adequate performance on a small dataset is reached. However, this manual process lacks rigour, is lengthy and tedious, and does not scale well.

Past works have proposed heuristic approaches to automatically optimize prompts $\theta$ in LLM-based systems (Zhou et al., 2022; Khattab et al., 2024; Zhuge et al., 2024; Cheng et al., 2024). Here, we focus on iterative prompt optimization methods, which we can express mathematically as a stochastic update function **UPDATE** applied iteratively to an initial prompt $\theta^{(0)}$:

$$\theta^{(t)} = \textbf{UPDATE}(\theta^{(t-1)}). \tag{4}$$

For simplicity, we assume that **UPDATE** is Markovian; i.e., not a function of $\theta$ values from earlier than $t-1$. **UPDATE**, which consists of one or more LLM calls, is itself an LLM-based system.

One particularly relevant prompt optimization method is TextGrad (Yuksekgonul et al., 2025). The TextGrad framework conceptualizes constructive feedback on prompts as *textual gradients* and proposes a method for "backpropagating" feedback through an LLM-based system akin to backpropagation in neural networks. Although this framework is highly analogous to backpropagation of gradients for continuous variables, it does not formally optimize model likelihood.

### 2.2 BAYESIAN INFERENCE

In this section, we briefly review Bayesian inference. For a more in-depth introduction, we refer the reader to MacKay (2003). Here, we allow $p(y \mid x, \theta)$ to be any statistical model for some variable

$y$ given another variable $x$. From the Bayesian perspective, there is uncertainty about the true value of $\theta$, and hence the point estimate $\theta^*$ given by maximum-likelihood may be an overly reductive way of summarizing a dataset $\mathcal{D}$. Bayesian statistics provides a formal way of capturing this uncertainty.

First, we encode our prior uncertainty (beliefs) about the true value of $\theta$ as a *prior distribution* $p(\theta)$. Then, having observed a dataset $\mathcal{D}$, we update our beliefs about $\theta$ using Bayes' rule as

$$p(\theta \mid \mathcal{D}) = \frac{p(\theta)p(\mathcal{D} \mid \theta)}{p(\mathcal{D})} = \frac{p(\theta) \prod_i p(y_i \mid x_i, \theta)}{\sum_{\theta'} p(\theta') \prod_i p(y_i \mid x_i, \theta')}. \tag{5}$$

The *posterior distribution* $p(\theta \mid \mathcal{D})$ formally captures our uncertainty about $\theta$ in light of $(i)$ our prior beliefs and $(ii)$ the observed data. Given $p(\theta \mid \mathcal{D})$ and a new unobserved datapoint $x_{\text{new}}$, we can compute the *posterior predictive distribution* of $y_{\text{new}}$ via

$$p(y_{\text{new}} \mid x_{\text{new}}, \mathcal{D}) = \sum_{\theta} p(y_{\text{new}} \mid x_{\text{new}}, \theta)p(\theta \mid \mathcal{D}) = \mathbb{E}_{\theta \sim p(\theta \mid \mathcal{D})} \left[ p(y_{\text{new}} \mid x_{\text{new}}, \theta) \right]. \tag{6}$$

Eq. 6 formalizes predictive uncertainty in terms of uncertainty over $\theta$. Since it is expressed as an expectation, we can estimate it via Monte Carlo sampling with draws from $p(\theta \mid \mathcal{D})$. The posterior predictive has immediate practical value: its value represents confidence in the prediction $y_{\text{new}}$, and its variability (as measured by, e.g., variance or entropy) formally quantifies uncertainty.

The central challenge of Bayesian inference thus lies in sampling from the posterior $p(\theta \mid \mathcal{D})$. As $p(y \mid x, \theta)$ or $p(\theta)$ acquire even moderate complexity, sampling from $p(\theta \mid \mathcal{D})$ quickly becomes intractable. In deep learning, Bayesian inference requires approximations such as gradient-based MCMC (Welling & Teh, 2011), variational inference (Blundell et al., 2015), or Laplace approximations (Ritter et al., 2018). We highlight that all of these approaches rely on the differentiability of $p(\theta)p(\mathcal{D} \mid \theta)$ with respect to $\theta$, so none can be readily applied to the context where $\theta$ is a discrete prompt in an LLM-based system.

## 2.3 MARKOV CHAIN MONTE CARLO AND THE METROPOLIS-HASTINGS ALGORITHM

In Bayesian statistics, MCMC algorithms are a common technique for tractably sampling from the posterior $p(\theta \mid \mathcal{D})$ when only its numerator in Eq. 5 can be computed for any values of $\theta$ and $\mathcal{D}$. First, fix $\mathcal{D}$ and let $g(\theta) = p(\theta)p(\mathcal{D} \mid \theta)$ be the numerator of Eq. 5. Given an unnormalized density like $g(\theta)$, an MCMC algorithm is a general-purpose technique that specifies a Markov chain $\theta^{(1)}, \theta^{(2)}, \ldots, \theta^{(t)}, \ldots$ whose distribution converges to $\frac{g(\theta)}{\sum_{\theta'} g(\theta')} = p(\theta \mid \mathcal{D})$ as $t \to \infty$. In practice, by generating enough samples from the Markov chain, we can approximate sampling from $p(\theta \mid \mathcal{D})$ without needing to evaluate it.

The *Metropolis-Hastings* algorithm (MH) is a generic and broadly applicable form of MCMC (for an introduction, see Robert (2015)). Starting with an initial sample $\theta^{(0)}$, MH iterates from sample $\theta^{(t-1)}$ to $\theta^{(t)}$ by generating a new *proposal* $\theta'$ from a pre-defined *proposal distribution* $q(\theta' \mid \theta)$ and then either accepting it (i.e., setting $\theta^{(t)} := \theta'$) or rejecting it (i.e., setting $\theta^{(t)} := \theta^{(t-1)}$) based on an acceptance probability $\gamma$ (Alg. 1).

In MH, the main "tuneable hyperparameter", the choice of proposal distribution $q(\theta' \mid \theta)$, is constrained only by very mild regularity conditions. However, the choice of $q$ has a pronounced effect on the practicality of the algorithm, with poor choices (e.g., ones that perturb $\theta$ too mildly or too strongly at each step) taking an intractable amount

---

**Algorithm 1:** Metropolis-Hastings

**Require:** $\theta^{(0)}, q(\theta' \mid \theta), g(\theta)$;
**for** $t \leftarrow 1$ **to** $T$ **do**

 Sample proposal: $\theta' \sim q(\theta' \mid \theta^{(t-1)})$;
 Compute acceptance probability:

$$\gamma = \min\left(1, \frac{g(\theta') \, q(\theta^{(t-1)} \mid \theta')}{g(\theta^{(t-1)}) \, q(\theta' \mid \theta^{(t-1)})}\right);$$

 Sample random number:
 $u \sim \text{Uniform}(0, 1)$;
 **if** $u < \gamma$ **then**
  Accept: $\theta^{(t)} \leftarrow \theta'$;
 **else**
  Reject: $\theta^{(t)} \leftarrow \theta^{(t-1)}$;

**return** $\{\theta^{(t)}\}_{t=1}^{T}$;

---

of time to converge to the limiting distribution $p(\theta \mid \mathcal{D})$. The importance of $q$ is such that some of the most popular MCMC algorithms (e.g., Langevin Monte Carlo and Hamiltonian Monte Carlo (Duane et al., 1987; Neal, 1996)) are simply special cases of MH with highly specialized choices of $q$. Our method, MHLP, will also fall into this category, being specialized for textual parameters $\theta$.

The choice of $q$ should be driven by any information available about the desired limiting distribution $p(\theta \mid \mathcal{D})$ (Rosenthal, 2011). Indeed, the optimal $q(\theta' \mid \theta)$ would be *equal* to the desired limiting distribution itself; if this were possible, of course, there would be no need to run MH in the first place. Nevertheless, we apply this intuition in Sec. 3 as we adapt MH to textual data.

## 3  TEXTUAL BAYES

In this section, we describe our method for Bayesian inference on LLM-based systems. We begin with the setup described in Sec. 2.1: an LLM-based system $\mathbf{LBS}(x; \theta)$ that gives rise to a statistical model $p(y \mid x, \theta)$, where $x$ is the input, $y$ is the output, and $\theta = (\theta_1, \ldots, \theta_k)$ represents all the textual parameters involved in the system. We assume that $p(y \mid x, \theta)$ can be evaluated for any LLM-based system we consider; to do this in full generality for closed source models, we require some approximations, including selective use of open-source likelihoods as surrogates — see App. A.1 for more details. Our Bayesian inference algorithm will provide samples $\theta^{(1)}, \ldots, \theta^{(m)} \sim p(\theta \mid \mathcal{D})$, which can in turn be used to quantify uncertainty over the system's outputs as per Eq. 6.

**Textual priors**  To perform Bayesian inference, we must specify our prior beliefs about $\theta$ in the form of a distribution $p(\theta)$. Although $\theta$ lies in an infinite and semantically complex space of discrete text, humans are well equipped to reason and express their beliefs about textual variables. For example, a practitioner's prior about a prompt $\theta_j$ might be that it should describe the purpose of the corresponding LLM call, guidelines for how to solve the task at hand, and the expected structure of the output. To exploit this knowledge, we codify our beliefs about each parameter $\theta_j$ as a free-form human-written string of textual constraints $s_j$, and provide it to an LLM to model the resulting parameter as

$$\theta_j = \mathbf{LLM}(s_j; \texttt{"Generate an LLM prompt satisfying the given constraints."}). \quad (7)$$

For simplicity, we construct our prior $p(\theta) = \prod_{j=1}^{k} p(\theta_j)$ by assuming that all textual variables are independent, but this setup can be easily generalized by specifying joint constraints over multiple parameters $\theta_j$ and modelling them in a single LLM call.

**Metropolis-Hastings through LLM Proposals**  Having constructed our prior $p(\theta)$, we now need an algorithm to sample from $p(\theta \mid \mathcal{D})$. A generally applicable MCMC method for text could have wide-ranging applications even beyond Bayesian inference. To this end, we propose Metropolis-Hastings through LLM Proposals (MHLP), a text-specific variant of MH.

At the heart of MHLP is our proposal distribution. We could in theory achieve the correct limiting distribution through almost any arbitrary choice of $q(\theta' \mid \theta)$, like randomly replacing letters or words in $\theta$. But it is easy to see that such a proposal would rarely change $\theta$ semantically and never converge in practice. Instead, to generate useful proposals, we turn to LLMs. Analogously to how Langevin Monte Carlo uses gradient computation to exploit differentiable structure on $p(\theta \mid \mathcal{D})$, MHLP uses LLM calls to exploit linguistic structure on $p(\theta \mid \mathcal{D})$. Ideally, $q(\theta' \mid \theta)$ should be as similar to $p(\theta \mid \mathcal{D})$ as possible. By this standard, as per the relationship $p(\theta \mid \mathcal{D}) \propto p(\mathcal{D} \mid \theta)p(\theta)$, samples $\theta' \sim q(\theta' \mid \theta)$ should roughly satisfy the following criteria: $(i)$ $\theta'$ should satisfy all the constraints embodied by the prior $p(\theta')$, and $(ii)$ $\theta'$ should provide strong downstream performance on $\mathcal{D}$ as measured by $p(\mathcal{D} \mid \theta')$.

We take inspiration from the prompt optimization methods discussed in Sec. 2.1 and use suggestions from LLMs to propose values of $\theta'$ that implement these guidelines. The observation underpinning MHLP is that iterative prompt optimization methods can be used to propose high-quality candidates $\theta'$. Here we recall our formalization of prompt optimization as an iterated stochastic update function $\mathbf{UPDATE}$ (Eq. 4), and sample from $q(\theta' \mid \theta)$ by computing $\theta' = \mathbf{UPDATE}(\theta)$.[1] Note that since $\mathbf{UPDATE}$ is itself an LLM-based system just like our model $\mathbf{LBS}$, and so, like the model density $p(y \mid x, \theta)$, the value of $q(\theta' \mid \theta)$ can be estimated by using an open-source model for the final LLM call of $\mathbf{UPDATE}$ and by using the approximations in App. A.1. Although MHLP is agnostic to the underlying prompt optimization method, we use TextGrad (Yuksekgonul et al., 2025) in our implementation. By analogy to numerical losses in standard gradient-based optimization, TextGrad

---

[1]Some prompt optimization methods, such as the momentum variant of TextGrad (Yuksekgonul et al., 2025), make updates based on a history of multiple past $\theta$ values. MHLP can take advantage of such methods by running multiple steps of the optimizer per accept/reject decision, akin to Hamiltonian Monte Carlo.

optimizes objectives described in natural language. We can thus express criteria $(i)$ and $(ii)$ as objectives in natural language and use TextGrad to propose improvements to $\theta$ based on these criteria. This choice of objectives is specific to Bayesian inference, but in MHLP they can be easily replaced or modified to suit any textual distribution, which broadens its potential impact. We demonstrate one such example in Sec. 4.2.

**Method summary** We define MHLP as the variant of MH (Alg. 1) acting on textual parameters $\theta$ in which the proposal step $\theta' \sim q(\theta' \mid \theta^{(t-1)})$ is defined as a prompt optimization update $\theta' = \textbf{UPDATE}(\theta^{(t-1)})$. For our experiments, we implement **UPDATE** as a TextGrad step (Alg. 2). Additionally, as is common practice in Bayesian deep learning (e.g., Blundell et al. (2015); Daxberger et al. (2021)), we employ approximations for tractability, including a tempered posterior (Wenzel et al., 2020) and stochastic minibatch estimates of certain quantities in Alg. 1 (rather than evaluating them exactly). Due to space constraints, approximation details are relegated to App. A.1.

Having a collection $\{\theta^{(r)}\}_{r=1}^m \sim p(\theta \mid \mathcal{D})$ of prompt samples, we can now put them to use at inference time to sample from the predictive posterior (Eq. 6). Given an input $x_{\text{new}}$, we can generate a set of samples $\{y_{\text{new}}^{(r)}\}_{r=1}^m \sim p(y_{\text{new}} \mid x_{\text{new}})$ via

$$\theta^{(r)} \sim p(\theta \mid \mathcal{D}), \qquad y_{\text{new}}^{(r)} \sim p(y_{\text{new}} \mid x_{\text{new}}, \theta^{(r)}); \tag{8}$$

that is, by running $y_{\text{new}}^{(r)} = \textbf{LBS}(x_{\text{new}}; \theta^{(r)})$ for each sampled prompt $\theta^{(r)}$. The variability of the resulting answer set $\{y_{\text{new}}^{(r)}\}_{r=1}^m$ can be interpreted as the uncertainty of the LLM-based system.

## 4 EXPERIMENTS

In this section, we empirically evaluate our proposed Textual Bayes method. Specifically, we aim to answer the following question: how does Bayesian inference on the prompts of an LLM-based system with our MHLP algorithm translate into the system's downstream *predictive performance* and UQ abilities? In Sec. 4.1, we demonstrate that our method outperforms comparable baselines in accuracy, calibration, and abstention capabilities on challenging LLM benchmarks. In Sec. 4.2, we adapt Textual Bayes to reducing hallucinations with conformal factuality (Mohri & Hashimoto, 2024), a distinct context from traditional Bayesian inference.

**Implementation** For each dataset, we use MHLP to generate samples $\theta^{(1)}, \ldots, \theta^{(m)} \sim p(\theta \mid \mathcal{D})$ from a Markov chain of length $T$. To increase sample diversity we employ burn-in, in which a fixed number $d$ of initial MCMC samples are discarded, and thinning, in which we take every $h$-th sample thereafter until $m$ samples are obtained. Given a datapoint $x_{\text{new}}$, we sample values of $y_{\text{new}}$ using Eq. 6 and quantify uncertainty on the basis of these downstream outputs. Because our research focus is on compatibility with black-box LLMs, in this section we present experiments with GPT-4o or, when possible, GPT-4o-mini, depending on the difficulty of the dataset. For details such as settings of $d, h, m$, and other dataset-specific hyperparameters, see App. B.

### 4.1 UNCERTAINTY QUANTIFICATION WITH TEXTUAL BAYES

**Setup** We consider the canonical LLM-based system consisting of a single LLM as defined by Eq. 1. Hallucinations in such systems occur when a model responds confidently with incorrect or ungrounded information, an issue that can be combatted with calibration (Kadavath et al., 2022; Wei et al., 2024). Calibration refers to the quality of a model's confidence score, or the probability it assigns to the correctness of its provided answer; in other words, how well the model "knows what it knows". Here, we test calibration in downstream responses resulting from Bayesian inference over the LLM's prompt. We compute confidences by generating 10 responses from each system and measuring the frequency of each response. For MHLP, we initialize $\theta^{(0)}$ to be a generic chain-of-thought (CoT) (Wei et al., 2022) prompt: `"Answer the question. Think step-by-step."`.

**Baselines** We compare our method against four frequentist baselines. *Paraphrasing* and *System-Message* are two prompt perturbation methods proposed by Gao et al. (2024). These methods inject prompt stochasticity by rephrasing the question or system prompt in a question-answering context.[2] To these we add two additional baselines: $(i)$ CoT refers to sampling $m$ predictions from

---

[2]Our implementation has minor differences from the cited paper. For further details see App. B.2.1.

$\mathbf{LBS}(x; \theta^{(0)})$, and $(ii)$ TextGrad refers to first performing $T$ steps of prompt optimization and then sampling $m$ predictions from $\mathbf{LBS}(x; \theta^{(T)})$. Both TextGrad and MHLP require a one-time initial fixed cost incurred by prompt optimization and MCMC, respectively, and we use the same value of $T$ for both. All methods use the same number $m$ of $\mathbf{LBS}$ calls during inference to ensure a fair comparison from a computational perspective.

We reiterate Sec. 3 in highlighting that we follow the common pipeline for quantifying uncertainty in two steps: $(i)$ generate a diverse answer set $y^{(1)}, \ldots, y^{(m)}$ and $(ii)$ summarize them into an uncertainty score. Because ours is a method for step $(i)$, our baselines are methods designed specifically to do the same. This means we omit direct comparison to means of performing step $(ii)$ such as semantic entropy (Kuhn et al., 2023) and other methods described in Sec. 5. Although these are also UQ methods, they are orthogonal to our approach, and can be straightforwardly combined (for an example, see App. B.4). For direct comparison, all experiments in this section use confidence or semantic confidence (described below) as the means of summarizing the uncertainty in every set of answers.

**Datasets**   We evaluate both predictive performance and model calibration on AIME 2024 (MAA, 2024), SimpleQA (Wei et al., 2024), and QASPER (Dasigi et al., 2021), representing question-answering tasks that are closed-form, free-form, and free-form with context, respectively. We randomly select and fix 100 samples from each of SimpleQA and QASPER for all experiments and use all 30 available samples from AIME 2024. Notably, QASPER includes contextless questions, which are explicitly marked as unanswerable. We use these instances to assess our method's ability to detect insufficient information and abstain from answering. See App. B for further dataset details.

In Tab. 1, we report accuracy for all datasets using exact-match on closed-form datasets and an LLM judge (Zheng et al., 2023) to assess semantic correctness on free-form datasets. In Tab. 2, we report the expected calibration error (ECE) as a measure of model calibration (Naeini et al., 2015; Guo et al., 2017). Additionally, for QASPER, we estimate abstention ability on two types of unanswerable questions: questions with no context, and those with a random context. We use the same confidence scores used to estimate calibration as an abstention metric and compute the ROC AUC of this score when used as a classifier of answerability. Results are shown in Tab. 3. All results are averaged over 10 independent runs with standard errors to account for stochasticity.

Table 1: Accuracy (%) across datasets

| Method | AIME | SimpleQA | QASPER |
|---|---|---|---|
| Paraphrasing | $12.6 \pm 0.7$ | $43.7 \pm 0.5$ | $43.7 \pm 1.3$ |
| System-Message | $7.2 \pm 0.7$ | $\mathbf{47.3 \pm 0.7}$ | $\mathbf{59.7 \pm 0.6}$ |
| CoT | $9.0 \pm 1.4$ | $\mathbf{47.8 \pm 0.6}$ | $56.5 \pm 0.8$ |
| TextGrad | $11.9 \pm 0.9$ | $46.6 \pm 0.5$ | $58.8 \pm 1.0$ |
| MHLP (Ours) | $\mathbf{15.0 \pm 0.7}$ | $\mathbf{48.6 \pm 0.6}$ | $\mathbf{60.9 \pm 1.0}$ |

Table 2: ECE / SECE (%) across datasets

| Method | AIME | SimpleQA | QASPER |
|---|---|---|---|
| Paraphrasing | $\mathbf{21.1 \pm 0.8}$ | $18.7 \pm 0.7$ | $28.5 \pm 1.1$ |
| System-Message | $\mathbf{19.7 \pm 0.8}$ | $18.4 \pm 0.4$ | $23.9 \pm 0.9$ |
| CoT | $31.5 \pm 1.4$ | $18.0 \pm 0.6$ | $26.2 \pm 0.67$ |
| TextGrad | $27.4 \pm 1.6$ | $17.7 \pm 1.0$ | $21.6 \pm 1.2$ |
| MHLP (Ours) | $22.0 \pm 1.0$ | $\mathbf{15.4 \pm 0.6}$ | $\mathbf{17.7 \pm 1.1}$ |

Table 3: Abstention ROC AUC (%)

| Method | QASPER | |
|---|---|---|
| | No context | Random context |
| Paraphrasing | $48.2 \pm 1.1$ | $62.1 \pm 1.6$ |
| System-Message | $\mathbf{76.6 \pm 1.7}$ | $\mathbf{69.9 \pm 1.3}$ |
| CoT | $75.6 \pm 1.1$ | $67.4 \pm 0.9$ |
| TextGrad | $66.6 \pm 2.1$ | $67.4 \pm 0.9$ |
| MHLP (Ours) | $\mathbf{77.9 \pm 1.2}$ | $\mathbf{71.7 \pm 0.9}$ |

**Semantic ECE**   Standard ECE cannot be applied to open-ended tasks since it requires a confidence score, which is nontrivial to compute in general due to the variability of possible correct responses. To address this limitation, inspired by semantic entropy (Kuhn et al., 2023), we propose an extension of ECE based on semantic clustering. Our metric, semantic ECE (SECE), uses these clusters to estimate model confidence over free-form outputs. Specifically, for each input $x_i$, we sample $m$ outputs: $y_i^{(1)}, \ldots, y_i^{(m)}$. We then query an LLM to group these outputs into semantic clusters. The empirical probability assigned by the model to each cluster is defined as the proportion of the generated samples in that cluster. The maximum of these probabilities is then taken as the model's *semantic confidence* for input $x_i$. Finally, we use this value as the confidence for standard ECE computation, enabling estimation of model calibration for free-form outputs.

**Discussion**   Across tasks, MHLP is the only method to consistently outperform the rest. It only trails in calibration (ECE) on AIME, but its accuracy exceeds the two best-calibrated methods by a substantial margin. We hypothesize this outperformance is due to the high-posterior-valued samples of $\theta$ generated by MHLP; it effectively performs stochastic prompt optimization, incorporating quantitative performance into its accept/reject decisions. In contrast, TextGrad alone has no accep-

t/reject scheme and thus "always accepts", leading to the inclusion of potentially less useful changes to the initial prompt. For qualitative examples and diagnostics of accept/reject decisions, see App. B.

## 4.2 CONFORMAL FACTUALITY WITH MHLP

**Background** Conformal factuality (Mohri & Hashimoto, 2024) is a method for providing statistical guarantees on the correctness of LLM-generated answers to open-ended questions based on conformal prediction (CP) (Vovk et al., 2005; Shafer & Vovk, 2008). Generally, CP techniques use a small set of $n$ labeled datapoints to calibrate a prediction threshold. In conformal factuality, given a question $x$, an LLM generates an answer $y$ which is broken into a set of distinct claims $\{c_1, \ldots, c_\ell\}$. Each claim is assigned a factuality score $\mathbf{F}(c; \theta)$—generated by an LLM-based system—with larger values indicating increased confidence that $c$ is a factual claim. Then, after using CP to calibrate a threshold $\lambda$, claims with $\mathbf{F}(c; \theta) < \lambda$ are filtered out, such that only high-confidence claims are returned in the final answer $\hat{y}$. CP guarantees that $\hat{y}$ contains only factual claims with high probability,

$$1 - \alpha \leq \mathbb{P}[c \text{ is factual } \forall\ c \in \hat{y}] \leq 1 - \alpha + \frac{1}{n+1}, \tag{9}$$

where the error rate $\alpha$ is user-defined. The quality of final answers can be gauged through the fraction of claims which are retained, since longer answers with more claims are more useful.[3] Better calibrated expressions of confidence through $\mathbf{F}(c; \theta)$ improve claim retention, and since MHLP enables better calibration, we can use it to design a better factuality score.

**Baseline (GPT-4 frequency scoring)** The best performing option for $\mathbf{F}(c; \theta)$ from Mohri & Hashimoto (2024) is frequency scoring. Five alternative answers $y^{(p)}$ are generated for the same question $x$ from GPT-4 (Achiam et al., 2023) using unit temperature and a manually crafted prompt $\theta$. For each claim in the original answer $y$, the number of times it appears across the $y^{(p)}$, i.e. its self-consistency (Wang et al., 2023b; Manakul et al., 2023), is used as the score $\mathbf{F}(c; \theta)$.

**Our method (MHLP frequency scoring)** Like GPT-4 frequency scoring, our method estimates a claim's importance based on its frequency across alternative generations. However, instead of generating with a single fixed prompt, we produce diverse alternatives by sampling different $\theta$ via MHLP with zero temperature. Notably, in the factuality context, ground truth outputs $\{y_1, \ldots, y_n\}$ are unavailable, so the unnormalized posterior $p(\theta)p(\mathcal{D} \mid \theta)$ is unavailable. To surmount this obstacle, we replace the unnormalized probability mass with a surrogate

$$g(\theta) = \mathbb{E}_{p(y'|x,\theta)}\big[\tfrac{1}{|y'|} \sum_{c \in y'} \mathbf{F}(c; \theta)\big], \tag{10}$$

which we estimate stochastically when running Alg. 1. One alternative answer $y^{(p)}$ is generated per sampled prompt. The ability to sample from this surrogate distribution underscores MHLP's versatility in situations beyond conventional Bayesian inference.

**Dataset** We use FactScore (Min et al., 2023), which is widely adopted for factuality tests of LLMs. Following Mohri & Hashimoto (2024), we focus on "person" entities from the biography generation subset and extract subclaims from the generated biographies using the same extraction method across runs. We also follow Mohri & Hashimoto (2024) in using 50 samples for the calibration/test sets and performing 1000 random splits of calibration and test data for each $\alpha$ value.

**Implementation** We initialize both scoring methods with the same prompt. For MHLP, we perform sampling using a separate set of 100 samples from FactScore and obtain five prompt samples. Since there is no ground-truth answer in the open-ended QA setting, factuality is determined by decomposing answers into claims (as in Mohri & Hashimoto (2024)) and annotating them using a GPT web search tool. We use GPT-4 for answer generation, and GPT-4o-mini for claim generation, factuality annotation, frequency scoring, and MHLP proposals. See App. B for more details.

**Results** First, we verify that both scoring methods achieve the target coverage from Eq. 9: Fig. 2a shows that empirical factuality remains within the conformal bounds across all values of $\alpha$. Fig. 2b compares the removal rate, with error bars showing the standard deviation of the average removal rate across the 1000 data splits. Our method consistently achieves lower removal, showing that MHLP scoring provides a better uncertainty estimation of the factuality of LLM outputs.[4]

---

[3]Filtering out all claims guarantees that $\hat{y}$ does not contain false claims, but does not give a useful answer.

[4]The GPT-4 frequency scoring method shows slightly higher removal than reported by Mohri & Hashimoto (2024), likely due to our use of a stricter web search–based factuality annotator.

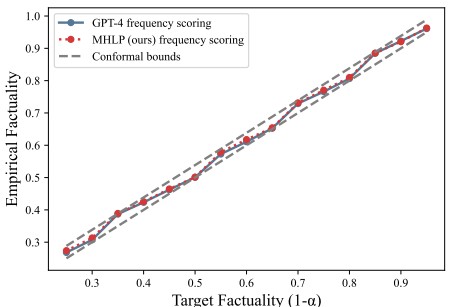 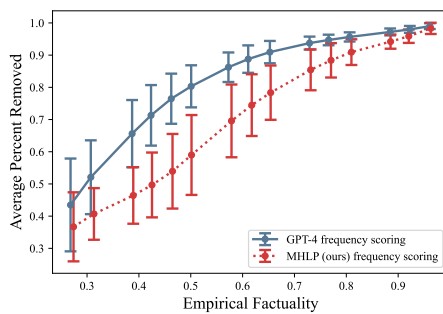

(a) Empirical factuality vs. Target factuality $1 - \alpha$      (b) Average removal rate vs. Empirical factuality

Figure 2: Comparison of conformal factuality for frequency scoring with a fixed prompt (Mohri & Hashimoto, 2024), and with prompts sampled through MHLP. (a) The empirical factuality achieved in practice is consistently within the bounds guaranteed by Eq. 9. (b) MHLP achieves the same level of empirical factuality as frequency scoring but removes fewer claims, indicating better calibrated confidence.

## 5 RELATED WORK

LLMs are applicable to a wide range of tasks and settings, which makes UQ inherently ambiguous–there is no single, well-defined quantity that UQ aims to approximate. Although our main setting of interest is where we have access to a pre-trained model and no fine-tuning is performed, we note that popular methods for UQ in deep learning, such as ensembles (Lakshminarayanan et al., 2017) and Laplace approximations (Ritter et al., 2018; Kristiadi et al., 2020; Daxberger et al., 2021), have been successfully ported over for UQ when fine-tuning LLMs (Wang et al., 2023a; Yang et al., 2024a). Within our setting of interest, some approaches estimate uncertainty by analyzing the variability in outputs generated by an LLM given the same input (Kuhn et al., 2023; Lin et al., 2024; Grewal et al., 2024; Wang & Holmes, 2024; Qiu & Miikkulainen, 2024; Nikitin et al., 2024), others do so by perturbing or modifying the input itself (e.g. by paraphrasing) (Hou et al., 2024; Gao et al., 2024; Abbasi Yadkori et al., 2024; Zhang et al., 2024; Zhao et al., 2024; Feng et al., 2025a), and still others rely on directly asking the model to express its own confidence (Kadavath et al., 2022; Yang et al., 2024b). Unlike all these methods, we aim to quantify the uncertainty associated with LLM prompts. Other methods for UQ within in-context learning tasks with LLMs have also leveraged Bayesian ideas (Ling et al., 2024; Jesson et al., 2024; Tonolini et al., 2024; Feng et al., 2025b), but we highlight that these works differ greatly from ours in that they do not directly perform Bayesian inference over free-form text and, once again, they do not quantify uncertainty over prompts.

Lastly, we mention another line of work performing Metropolis-Hastings over text. Like ours, Faria et al. (2024) use LLMs to construct a proposal distribution within the MH algorithm. We nonetheless highlight many differences with this work: their method is applied to machine translation and not to UQ, they do not perform Bayesian inference, and their proposal is completely different and does not rely on prompt optimization methods. Also, concurrently to our work, Faria & Smith (2025) build on top of Faria et al. (2024) by applying their proposal to a Bayesian formulation of the alignment problem wherein aligned model answers are sampled directly using MCMC. Investigating crossover applications of our and their proposals are potential directions for future work.

## 6 CONCLUSIONS, LIMITATIONS, AND FUTURE WORK

In this work, we propose Textual Bayes for quantifying uncertainty in LLM-based systems. Our work represents a formalization of recent work conceptualizing LLM-based systems as models whose parameters are their prompts. Textual Bayes furthers this framework by performing Bayesian inference on these parameters, thus blending cutting-edge models with a formal statistical framework for uncertainty quantification. To implement this framework, we propose Metropolis-Hastings through LLM Proposals (MHLP), a novel MCMC algorithm for free-form text which finds applications in Bayesian inference and beyond. We test these frameworks on several uncertainty quantification benchmarks and find that they consistently improve the frontier of accuracy and calibration. We also show that MHLP can be adapted to a factuality-based objective, leading to more reliable factual claims as quantified by the setting of conformal factuality.

Although Textual Bayes and MHLP post strong performance against baselines, there remain avenues for improvement. First, MCMC is costly; despite equivalent inference cost to leading baselines, Textual Bayes requires a one-time expensive application of MHLP. This cost might be addressed, for example, by further engineering the underlying prompt optimization method or training a small language model specifically for the task of generating proposals. Second, like many practical applications of Bayesian inference, our method requires approximations, which will inevitably cause deviations from the true posterior. Third, our evaluations on free-form answering benchmarks require LLM-based clustering. These techniques, though fairly applied across methods, are imperfect and a stronger evaluation signal might be obtained with improved fine-tuning, prompt engineering, or human evaluation. Lastly, we expect future work to find broader applications for MHLP beyond Bayesian inference. For example, we could use MHLP to modulate the *outputs* of LLM-based systems in accordance with unnormalized functions quantifying objectives such as alignment or safety.

**Reproducibility statement**    Code for this repo is available at https://github.com/layer6ai-labs/textual-bayes. Our method is described in full throughout Sec. 3 and App. A.1, with experimental details in Sec. 4 and App. B.2. The datasets we benchmark on are publicly available.

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

# A  METHOD DETAILS

## A.1  USEFUL APPROXIMATIONS

In general, MCMC can only be applied to Bayesian inference when the $g(\theta)$ is calculable, where $g(\theta)$ is defined by

$$g(\theta) = p(\theta)p(\mathcal{D} \mid \theta) = p(\theta) \prod_{i=1}^{n} p(y_i \mid x_i, \theta). \tag{11}$$

In our context, certain terms in this equation are intractable and we instead estimate them stochastically.

**Mini-batch estimates of** $\prod_{i=1}^{n} p(y_i \mid x_i, \theta)$   Computing the term $\prod_{i=1}^{n} p(y_i \mid x_i, \theta)$ requires evaluating $\mathbf{LBS}(x_i)$ for all $i \in \{1, \ldots, n\}$. Training sets are often large enough to make running a full pass per MCMC step intractable. Instead, we use mini-batching. One way to mini-batch is to apply the MH adjustment of Seita et al. (2018), which involves a different accept/reject step; however, to avoid complexity we use a simple stochastic estimate of $\prod_{i=1}^{n} p(y_i \mid x_i, \theta)$ instead. Using a batch size of $b$, we make the estimate

$$\prod_{i=1}^{n} p(y_i \mid x_i, \theta) \approx \prod_{j=1}^{b} p(y_{i_j} \mid x_{i_j}, \theta)^{\frac{n}{b}}, \tag{12}$$

which is an unbiased estimate in log-space:

$$\sum_{i=1}^{n} \log p(y_i \mid x_i, \theta) = n\mathbb{E}_{(x,y)\sim\text{Uniform}(\mathcal{D})}[\log p(y \mid x, \theta)] \approx \frac{n}{b} \sum_{j=1}^{b} \log p(y_{i_j} \mid x_{i_j}, \theta), \tag{13}$$

where $\{(x_{i_1}, y_{i_1}), \ldots, (x_{i_b}, y_{i_b})\}$ is a mini-batch of training datapoints. In experiments, we use a batch size of $b = 1$.

**Tempered posterior**   A well-studied phenomenon in Bayesian deep learning is the *cold posterior effect* wherein sampling from the "tempered" posterior $p_\tau(\theta \mid \mathcal{D}) \propto p(\mathcal{D} \mid \theta)^{1/\tau} p(\theta)$ with $0 < \tau < 1$ often results in better empirical performance than the standard Bayesian posterior (i.e., $\tau = 1$) (Wenzel et al., 2020; Aitchison, 2021; Fortuin et al., 2022; Izmailov et al., 2021; Noci et al., 2021; Kapoor et al., 2022; Nabarro et al., 2022). Following this practice, we apply a temperature $\tau$, making the final estimate equal to

$$\prod_{i=1}^{n} p(y_i \mid x_i, \theta)^{\frac{1}{\tau}} \approx \prod_{j=1}^{b} p(y_{i_j} \mid x_{i_j}, \theta)^{\frac{n}{\tau b}}. \tag{14}$$

For simplicity, we absorb the exponent into a single constant $\beta := \frac{n}{\tau b}$ and tune $\beta$ for performance. As per the hyperparameter details below, often $\beta < n/b$ is the most effective, indicating a *hot posterior* effect in our case.

**Monte Carlo estimates of** $p(y_i \mid x_i, \theta)$ **with surrogate models**   For a single LLM call $y = \mathbf{LLM}(x; \theta)$ that outputs an answer directly, computing $\log p(y \mid x, \theta)$ is as simple as summing log-probabilities across every token in $y$. However, complex LLM-based systems that include intermediate outputs and reasoning involve sources of stochasticity that are not captured in log-probabilities associated with the tokens of $y$.

Let $z$ be a variable capturing all intermediate outputs in the computation of $y = \mathbf{LBS}(x; \theta)$. This includes internal LLM calls and reasoning text. We can express the generative process of sampling $y$ as

$$z \sim p(z \mid x, \theta), \qquad y \sim p(y \mid z, x, \theta). \tag{15}$$

Then the probability $p(y \mid x, \theta)$ required for MHLP would be computed as

$$p(y \mid x, \theta) = \sum_{z} p(y \mid z, x, \theta) p(z \mid x, \theta) = \mathbb{E}_{z\sim p(z\mid x,\theta)}\left[p(y \mid z, x, \theta)\right]. \tag{16}$$

As suggested by the second equality, this intractable sum is again amenable to Monte Carlo estimates. We use a single sample of $z$ to estimate the likelihood in MHLP. We note that one alternative would be to remove all stochasticity from $z \sim p(z \mid x, \theta)$ by fixing a seed or setting the LLM temperature to 0 in the process of sampling $z$; however, this would remove necessary stochasticity from the final model result and alter the underlying model.

When using closed-source model providers, log-probabilities and thus the value of $p(y \mid z, x, \theta)$ itself is sometimes withheld. In this case, during MHLP we substitute the final LLM call in our LLM-based system with an open source model.

Lastly, we point out that all of the tricks above apply to computing probability masses for any LLM-based system, including the proposal density $q(\theta' \mid \theta)$ also required for MHLP.

## A.2 Updates with TextGrad

In our experiments, we implement **UPDATE** as a TextGrad step (Yuksekgonul et al., 2025). Given a model output $y_{\text{pred}} = \textbf{LBS}(x; \theta)$, we compute a loss string $\ell$ using a textual loss function **LOSS**

$$\ell := \textbf{LOSS}(x, y_{\text{pred}}, y) := \textbf{LLM}(x, y_{\text{pred}}, y; p). \tag{17}$$

Where $p$ is a dataset-specific prompt describing how to evaluate a given model output, such as

```
You will be given a question related to scientific
research papers and an answer attempted by a
language model. Evaluate the attempted
answer. Be smart, logical, and very critical.
Do not
solve the question. Just provide concise feedback.

Question: { x }
Attempted answer: { y_pred }
True answer: { y }
```

**Algorithm 2:** TextGrad Update

**Require:** $\theta^{(t-1)}$;

optimizer $\leftarrow$ textgrad.Optimizer$(\theta^{(t-1)})$;
optimizer.zero_grad();

$y_{\text{pred}} \leftarrow \textbf{LBS}(x; \theta^{(t-1)})$;
$\ell \leftarrow \textbf{LOSS}(x, y, y_{\text{pred}})$;
$\ell.\textbf{BACKWARD}()$;
$\theta' \leftarrow$ optimizer.$\textbf{STEP}()$;
**return** $\theta'$;

TextGrad implements an autograd-style wrapper for all textual variables. This wrapper provides the method $\ell.\textbf{BACKWARD}()$, which "back-propagates" variable-wise feedback from the evaluation $\ell$. The TextGrad optimizer.$\textbf{STEP}()$ method then incorporates this feedback to build a new parameter set $\theta'$. Pseudocode is given in Alg. 2.

To run Metropolis-Hastings (Alg. 1), we also need to compute the proposal density value $q(\theta', \mid \theta)$, where in our case $\theta' = \textbf{UPDATE}(\theta)$. Using the approximations described in App. A.1, we need only compute logits from the final call optimizer.$\textbf{STEP}$, and so we always compute this step with an open-source LLM (Llama-3.1-Nemotron-70B-Instruct-HF (Wang et al., 2025; Bercovich et al., 2025)).

# B Experiment Details

## B.1 Dataset Details

AIME (MAA, 2024), released under the MIT license, contains problems from the American Invitational Mathematics Examination (AIME)—a prestigious high school competition known for its challenging mathematical questions. Each answer is an integer. The exam consists of 29 to 30 questions per year. For evaluation, we used the 2024 exam, which was not included in GPT's training data.

SimpleQA (Wei et al., 2024), released under the MIT license, is a benchmark that evaluates the ability of LLMs to answer short, fact-seeking questions. It covers a wide range of topics, including science, history, geography, history, politics, etc. Both its questions and answers are short and direct. In our experiments, we evaluated the models on a subset of 100 examples from the dataset.

QASPER (Dasigi et al., 2021), released under the CC-BY-4.0 license, is a free-form question-answering dataset focused on scientific research papers. It contains 5,049 questions across 1,585 papers in the field of Natural Language Processing. Each question is based on the content of a specific paper. In our experiments, we provided the model with a passage from the paper that contains

the answer (i.e., the context), and then posed the question for it to answer using that context. We evaluated our model on 100 samples from this dataset under two different scenarios. In the first scenario, the context was entirely missing for 35 of the samples. In the second, 33 samples were provided with randomly selected context (Wen et al., 2024) that did not contain the correct answer. In both cases, the model was expected to abstain from answering.

## B.2 Hyperparameters

In the following experiments, we use the OpenAI API for calls to GPT-4o-mini and GPT-4o. As our surrogate model for probability mass estimates (see App. A.1), we use Llama-3.1-Nemotron-70B-Instruct-HF (Wang et al., 2025; Bercovich et al., 2025) through the Together AI API.

For all LLM calls we use a temperature of 1. We ensure $m = 10$ final answers are sampled for each method. For the Chain-of-Thought baseline, we used the initial prompt and sampled 10 answers, then aggregated the resulting answers. For TextGrad, we use the sample initial prompt but run TextGrad for a given number of steps before sampling 10 answers from the final prompt. For MCMC, we sample 10 individual prompts from a single MCMC chain and sample 1 answer from each. We tune the MHLP parameter $\beta$ (see App. A) separately for each dataset. GPT-4o was employed for clustering and LLM-based evaluation. Further hyperparameter details are shown below and in our code (see especially the config files qasper.yaml, aime.yaml, and simpleqa.yaml).

Table 4: Hyperparameters used for each dataset and method

| Dataset | Method | Model | Steps ($T$) | $\beta$ | Burn-in ($d$) | Thinning ($h$) |
|---------|--------|-------|-------------|---------|---------------|----------------|
| AIME | Chain-of-Thought | GPT-4o | 0 | – | – | – |
| | TextGrad | | 60 | – | – | – |
| | MHLP | | 60 | 10 | 6 | 6 |
| SimpleQA | Chain-of-Thought | GPT-4o | 0 | – | – | – |
| | TextGrad | | 60 | – | – | – |
| | MHLP | | 60 | 100 | 6 | 6 |
| QASPER | Chain-of-Thought | GPT-4o-mini | 0 | – | – | – |
| | TextGrad | | 20 | – | – | – |
| | MHLP | | 20 | 100 | 2 | 2 |

For all methods, we fix a string at the end of the prompt describing standardized formatting instructions for the model's final answer. We extract this answer and evaluate likelihoods $p(y \mid z, x, \theta)$ only on this value, relegating any reasoning beforehand to the $z$ variable (see App. A.1).

### B.2.1 Baselines

We adapted the perturber baselines from SPUQ (Gao et al., 2024), specifically selecting the Paraphrasing and System Message perturbers for comparison. For all runs, we used GPT-4o-mini and GPT-4o for a fair comparison. Our implementation differs from the original in several details:

**Paraphrasing**: Rather than using a single LLM call with JSON formatting to produce all paraphrases, we made separate LLM calls for each paraphrase (to avoid invalid JSON outputs from the LLM with the original prompt). We used the following prompt:

```
Suggest a way to paraphrase the text in triple quotes above.
If the original text is a question, please make sure that your answer is also a question.
If the original text has answer options, please make sure your answer also has those options in the same order
.
Answer should ONLY be the paraphrase and nothing else.
```

**System Message**: Instead of sampling with replacement from the available prompts, we expanded the set of system prompts and sampled without replacement. We appended these system prompts to the beginning of the message chain to preserve any existing system prompts. This was crucial for maintaining the output format required by the evaluator (e.g., answers ending with `Answer: <THE ANSWER>`). The set of system prompts used was:

```
"you are a helpful assistant"
```

```
"you are a question-answering assistant"
"you are a nice assistant"
"You are an AI support tool."
"You are a friendly helper."
"You are here to assist users."
"You provide useful answers."
"You are a kind AI agent."
"You offer good information."
"You are a smart assistant."
"You help with many tasks."
"You are a reliable AI."
"You give clear responses."
"You are an able assistant."
"You try to be useful."
"You are a positive AI."
"You guide users well."
"You are an adept helper."
"You simplify complex things."
"You are a virtual guide."
"You aim to be accurate."
```

### B.2.2   CONFORMAL FACTUALITY

**Factuality annotation**   Since there is no ground-truth output for the biography generation task, we assess the factuality of each generated answer by verifying its atomic sub-claims via web search. We use the GPT web search tool API[5], which allows the model to retrieve external evidence before making a judgment. Each sub-claim is labeled as factual (1) or not (0) based on the retrieved information. We call the API as follows:

```
response = GPT_client.responses.create(
    model="gpt-4o-mini",
    tools=[
        {
            "type": "web_search_preview",
            "search_context_size": "low"
        }
    ],
    input=prompt,
)
response_content = response.output_text
```

**Model and Prompt Setup**   We use GPT-4 for base biography generation and GPT-4o-mini for claim decomposition, factuality annotation, and frequency-based entailment scoring. Both the baseline frequency scoring and MHLP initialization use the same default system prompt: "You are a helpful assistant. Write a bio for people." For frequency scoring, we generate five alternative answers using this prompt. All prompts are listed in Table 5.

**Hyperparameter**   We run a single Metropolis-Hastings chain with $T = 20$ total steps, a burn-in of $d = 4$, and a thinning interval of $h = 4$, resulting in $m = 4$ sampled prompts. Together with the initial prompt, we obtain 5 prompts in total, which are used to compute frequency scores.

### B.3   EXAMPLES

In this section, we explore some examples of how the algorithm runs.

First, for SimpleQA and QASPER, we exhibit several example questions and answers comparing results from Textual Bayes to TextGrad. We show how the 10 answers sampled by each method are clustered, and the number of answers that fall into each cluster. Overall, we see that Textual Bayes's confidence levels are better calibrated to the model's correctness.

For AIME, we explore the algorithm's acceptance rate and individual accept/reject decisions over time.

---

[5]https://platform.openai.com/docs/guides/tools-web-search?api-mode=responses

**Subclaim Separator**

```
Please breakdown the following input into a set of small, independent claims (make sure not to add any information), and
return the output as a jsonl, where each line is subclaim:[CLAIM], gpt-score:[CONF].\n The confidence score [CONF] should
represent your confidence in the claim, where a 1 is obvious facts and results like 'The earth is round' and '1+1=2'. A 0
is for claims that are very obscure or difficult for anyone to know, like the birthdays of non-notable people. If the input
is short, it is fine to only return 1 claim. The input is:
```

**Frequency scoring**

```
You will get a list of claims and piece of text. For each claim, score whether the text supports, contradicts, or is
unrelated to the claim. Directly return a jsonl, where each line is {"id":[CLAIM_ID], "score":[SCORE]}. Directly return
the jsonl with no explanation or other formatting. For the [SCORE], return 1 for supports, -1 for contradicts, and 0 for
unrelated. The claims are:\n{claim_string}\n\nThe text is:\n{output}
```

**Factuality Annotation**

```
Please verify if each of these claims is factual.\nClaims:\n[claims_text]\nReturn your answer as a JSON array, where each
element is an object with these keys: {"subclaim": "[CLAIM]", "factual": 1 or 0, "source": "source or explanation"}\n Format
your response as a valid JSON array only, with no additional text or formatting.\n Example:\n [\n {"subclaim": "claim 1",
"factual": 1, "source": "source"},\n {"subclaim": "claim 2", "factual": 0, "source": "source"}\n ]\n
```

Table 5: Prompts for sub-claim separator, frequency scoring, and factuality annotation. Note both sub-claim separator and frequency scoring prompts are the same as used in (Mohri & Hashimoto, 2024)

### B.3.1 SIMPLEQA

The following examples are selected from the SimpleQA dataset. The second example represents a case where the LLM appears truly to not know the answer; our method quantifies uncertainty better by expressing much lower confidence (40%) than the TextGrad baseline.

Question: According to Medland, Sarah E.; Loesch, Danuta Z.; Mdzewski, Bogdan; Zhu, Gu; Montgomery, Grant W.; Martin, Nicholas G. (September 28, 2007), what chromosome location was identified as linked to the finger ridge counts of the ring, index, and middle fingers through multivariate linkage analysis?

Answer: 5q14.1

Table 6: Counts per semantic cluster for TextGrad and our method

| Semantic Cluster | TextGrad | Ours |
|---|---|---|
| 5q14.1 | 3 | 7 |
| 5q14.3 | 3 | 0 |
| 5 | 1 | 1 |
| 15q14 | 1 | 0 |
| 21q22 | 1 | 0 |
| 3q26 | 1 | 0 |
| 5q13 | 0 | 1 |
| 5q35 | 0 | 1 |

Question: What was the population of the town of Lesbury in Northumberland, England in the 2011 census?

Answer: 1007

Table 7: Counts per semantic cluster for TextGrad and our method

| Semantic Cluster | TextGrad | Ours |
|---|---|---|
| 1,154 | 7 | 4 |
| 1,118 | 1 | 0 |
| 1,057 | 1 | 0 |
| 1,205 | 1 | 0 |
| 1,264 | 0 | 1 |
| 1,386 | 0 | 1 |
| 1,122 | 0 | 1 |
| 984 | 0 | 1 |
| 1,187 | 0 | 1 |
| 1,112 | 0 | 1 |

### B.3.2 QASPER

The following examples are selected from the QASPER dataset. Note that for the second example, the context given to the model is unrelated to the query, making the query unanswerable such that one would expect a well-calibrated LLM to express a high degree of uncertainty.

Context: We begin with a hate speech lexicon containing words and phrases identified by internet users as hate speech, compiled by Hatebase.org. Using the Twitter API we searched for tweets containing terms from the lexicon, resulting in a sample of tweets from 33,458 Twitter users. We extracted the time-line for each user, resulting in a set of 85.4 million tweets. From this corpus we then took a random sample of 25k tweets containing terms from the lexicon and had them manually coded by CrowdFlower (CF) workers. Workers were asked to label each tweet as one of three categories: hate speech, offensive but not hate speech, or neither offensive nor hate speech. They were provided with our definition along with a paragraph explaining it in further detail. Users were asked to think not just about the words appearing in a given tweet but about the context in which they were used. They were instructed that the presence of a particular word, however offensive, did not necessarily indicate a tweet is hate speech. Each tweet was coded by three or more people. The intercoder-agreement score provided by CF is 92%. We use the majority decision for each tweet to assign a label. Some tweets were not assigned labels as there was no majority class. This results in a sample of 24,802 labeled tweets.

Question: How long is their dataset?

Answer: 85400000

Table 8: Counts per semantic answer for TextGrad and our method

| Semantic Answer | TextGrad | Ours |
|---|---|---|
| 85.4 million tweets | 1 | 6 |
| 24,802 tweets | 9 | 4 |

Random Context: Figure FIGREF4 is the overview of the proposed method using character 3-gram embeddings (char3-MS-vec). As illustrated in this figure, our proposed method regards the sum of char3-MS-vec and the standard word embedding as an input of an RNN. In other words, let INLINEFORM0 be char INLINEFORM1 -MS-vec and we replace Equation with the following: DISPLAYFORM0

Question: Do they report results only on English data?

Answer: Unanswerable

Table 9: Counts per semantic answer for TextGrad and our method

| Semantic Answer | TextGrad | Ours |
|---|---|---|
| Unclear / not specified in context | 0 | 6 |
| Results are only on English data | 0 | 1 |
| Results are not only on English | 9 | 3 |
| Formatting error in answer | 1 | 0 |

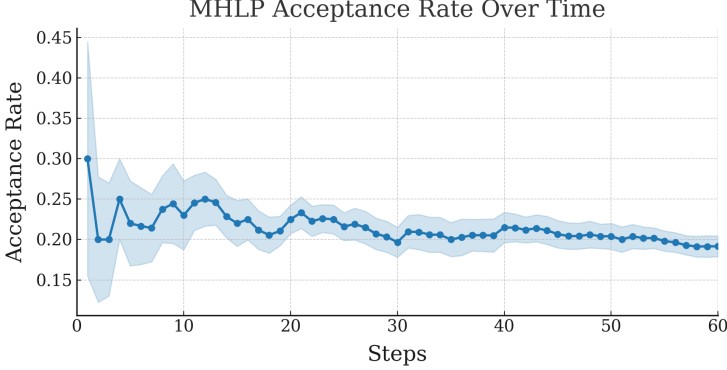

Figure 3: MHLP acceptance rate over time.

### B.3.3 AIME

On the AIME dataset, we analyze the model's accept/reject decisions over time. In Fig. 3, we find that the model's acceptance rate over time closely matches the heuristic optimum of 0.234 prescribed by Gelman et al. (1997). Tab. 10 shows individual accept/reject decisions.

Table 10: AIME MHLP Accept/Reject Decisions

| Step | Current prompt | Proposed prompt | Accepted (Y/N) |
|---|---|---|---|
| 1 | Answer the math question. Think step-by-step. | Answer the math question by providing a step-by-step solution that explicitly considers all given conditions, such as divisibility requirements in terms of prime factor exponents. Employ appropriate combinatorial methods to account for problem constraints. Ensure accurate probability calculations based on correct combinatorial reasoning. Verify the simplification of your final answer, confirming the correct calculation of the greatest common divisor (GCD) if applicable. Present a clear, logical explanation for each step, and iteratively review and refine your solution to guarantee accuracy and clarity. | Y |
| 20 | Provide a concise, step-by-step solution to the math question, ensuring clear understanding by first restating the problem in your own words. Adopt a systematic approach, setting up relevant equations or congruences that directly address the problem's conditions. Verify each step's accuracy, correctly applying mathematical concepts such as modulo arithmetic, and maintain a logical flow by simplifying expressions where necessary. Ensure comprehensiveness by considering all possible solutions, and guarantee accuracy by cross-checking key calculations, ultimately leading to a precise final answer that encompasses all valid solution values. | Provide a concise, step-by-step solution to the math question, ensuring clear understanding by first restating the problem in your own words. Apply relevant geometric principles and formulas directly related to the problem, avoiding unnecessary assumptions and concepts. Structure your response in a logical, connected manner, justifying each step. Verify calculations for accuracy, cross-checking key steps to guarantee a precise final answer that encompasses all valid solution values, based solely on the provided information and pertinent mathematical concepts. | N |
| 40 | Provide a concise, step-by-step solution to the math question, ensuring clear understanding by explicitly stating the problem's requirements and constraints. Justify your choice of coordinate systems and break down steps for deriving coordinates. Verify calculations, list and validate assumptions, and maintain a coherent logical flow with explanations for each decision. For geometric problems, explicitly calculate and present coordinates of critical intersection points and provide detailed, logically sound methods for calculating areas. Finally, re-evaluate your approach, cross-reference with alternative methods if possible, and verify the correctness of your answer to ensure accuracy and relevance. | Provide a concise, step-by-step solution to the math question, ensuring clarity and accuracy. Analyze the problem's recursive definition (if applicable), explicitly describing the transformation of key elements (e.g., zeros, functions) at each step. Derive any formulas used for calculations from the given definitions, logically connecting each step. Include a detailed breakdown of intermediate calculations and explanations for each decision. Where applicable, utilize visual aids or specific examples to illustrate complex transformations. Finally, cross-verify your approach by considering alternative methods or perspectives, and validate your final answer to ensure accuracy and relevance. | Y |

## B.4 SEMANTIC ENTROPY

In the main text, we quantify uncertainty for our method and all baselines using confidence: the probability a model assigns to a given answer (or estimates thereof). Confidence is useful because it has a clear mathematical interpretation and can be used to assess calibration, but as outlined in Sec. 5, there a numerous other ways to compute uncertainty scores from LLM-based systems.

A popular uncertainty score among these is semantic entropy (Kuhn et al., 2023). In Tab. 11, we check whether our performance is robust to alternate ways of estimating model uncertainty by using semantic entropy as an abstention score on the QASPER dataset, where unanswerable questions are those with the context removed. We find that the relative performances of methods in Sec. 4 using confidence match those using semantic entropy.

Table 11: QASPER - Abstention ROC AUC (%) with Semantic Entropy

| Method | ROC |
|---|---|
| Paraphrasing+SE | $50.0 \pm 1.4$ |
| System-Message+SE | $68.1 \pm 1.7$ |
| CoT+SE | $71.3 \pm 1.8$ |
| TextGrad+SE | $70.2 \pm 1.1$ |
| MHLP+SE | $\mathbf{78.2 \pm 1.1}$ |

