# OpenReview forum: "Textual Bayes: Quantifying Prompt Uncertainty in LLM-Based Systems"
_ICLR.cc/2026/Conference — ICLR 2026 Poster_

### Official Review · Reviewer_j29K · 2025-10-26

**Soundness:** 3
**Presentation:** 2
**Contribution:** 2
**Rating:** 4
**Confidence:** 4

**Summary:**

The paper proposes to see the prompts as "textual variables" for LLMs.\
They approximate the posterior distribution of the textual variable with MH technique.\
In MH technique, prior is defined with human constraints, and proposal distribution is defined as Update, which is TextGrad in this paper.\
After this process, we can sample multiple textual variables (i.e., prompts) from the posterior distribution.\
Lastly, by using these multiple prompts, we can approximate the posterior predictive distribution.

**Strengths:**

1. The paper is well-motivated.
- The large language models indeed show some vulnerability depending on the prompt, and prompt engineering is widely adopted.

2. The proposed method to approximate the posterior distribution is sound and solid.
- MH algorithm is a widely adopted technique to sample from an arbitrary distribution.
- The textual prior defined in Eq.7 is understandable.
- The proposal distribution defined with TextGrad is very interesting and makes sense for me.

**Weaknesses:**

1. Bayesian lens
- The title of this paper is Title: "TEXTUAL BAYES: QUANTIFYING UNCERTAINTY IN LLM-BASED SYSTEMS".\
However, in my opinion, the paper only deals with "prompt uncertainty"
- A true Bayesian lens would be the posterior over "model parameter", rather than over textual variables.\
For example, if a user inputs the question with his own prompt, then this method cannot quantify the uncertainty.\
In this sense, I think the prompt is more of an input rather than a parameter.


2. "To the best of our knowledge, we are the first both to quantify the uncertainty associated with prompts in LLM-based systems." could be an over-claiming.
- [A] and [B] also quantify the uncertainty associated with prompts in LLM-based systems.\
Indeed, [B] defines "prompt uncertainty" in their paper, and tries to quantify it.
- [A] Uncertainty Quantification with Pre-trained Language Models: A Large-Scale Empirical Analysis, EMNLP'22
- [B] Uncertainty Quantification and Decomposition for LLM-based Recommendation, WWW'25


3. Method section
- It ends with samples ($\theta$). What's next?
- It would be better to link how those samples can be utilized for the posterior predictive distribution in Eq.6.


4. Experiment
-  A direct evaluation of "uncertainty" of the predictive distribution (Eq.6) would be needed.\
We can compute the entropy of Eq.6 and adopt metrics like Kendall's tau to evaluate how well the uncertainty is related to the performance.
- I think the calibration is more like a post-processing process.

**Questions:**

1. Is there any warm-up steps for prompt sampling?\
In MH, by TextGrad, the prompt is gradually adapted to the true distribution.

2. $\\mathcal{D}$ is a single sample or an entire dataset?\
I think it should be an entire dataset.\
Can you elaborate more on how TextGrad updates a single prompt for the entire dataset?

---

> ### Author Response · Authors · 2025-11-22
>
> Thank you for spending your time and effort reviewing our work. We are happy to see that you find our proposed method sound and well-motivated. Our responses to your concerns and questions are below. If you feel they have been addressed, we kindly ask you to consider raising your score.
>
>
> > The title of this paper is .. "TEXTUAL BAYES: QUANTIFYING UNCERTAINTY IN LLM-BASED SYSTEMS". However, in my opinion, the paper only deals with "prompt uncertainty"
>
> We agree with this concern, which was shared by reviewer **jbfF**. The main novelty of our paper is indeed in how we quantify uncertainty over prompts in LLM-based systems, so to emphasize this we will adjust the title to “Textual Bayes: Quantifying *Prompt* Uncertainty in LLM-Based Systems”.
>
> > A true Bayesian lens would be the posterior over "model parameter", rather than over textual variables.
>
> You are correct that “model parameters” (i.e., weights) are typically the subject of Bayesian inference in deep learning, but it is completely mathematically valid to think of a prompt as a parameter in the Bayesian sense. Any quantity on which the observables' distribution depends can be defined as a Bayesian parameter. When working with blackbox LLMs, users do not have control over the model weights, only the ability to adjust prompts to shape the output distribution of the underlying model. Hence, our method to quantify uncertainty on these textual parameters is all the more important in this context.
>
> > For example, if a user inputs the question with his own prompt, then this method cannot quantify the uncertainty. In this sense, I think the prompt is more of an input rather than a parameter.
>
> It is true that an LLM user can technically craft their own input prompt. However, here we focus on the typical case, where the entire prompt given to an LLM is broken into multiple parts: the system prompt and the user prompt/query, which are concatenated. Since the system prompt is typically fixed and reused for a given task (a set of input queries), it certainly is a tunable parameter of the model. Our method quantifies uncertainty over this textual parameter, which is separate from the user query, which is certainly an input.
>
> > "To the best of our knowledge, we are the first both to quantify the uncertainty associated with prompts in LLM-based systems." could be an over-claiming.
>
> Thank you for bringing up these sources—we were unaware of both and will cite them in the revision. We agree with your comment, and will tone down this claim to the following:
>
> 	“To the best of our knowledge, we are the first to *perform Bayesian inference* over the space of free-form prompts in LLM-based systems.”
>
> > It ends with samples ($\theta$). What’s next? It would be better to link how those samples can be utilized for the posterior predictive distribution in Eq.6.
>
> We appreciate the suggestion here—the downstream use of these samples could certainly be made more clear at the end of the method section. We will add some additional clarity in the revision leading into the experimental section. This connects back to Eq. 6, which can be estimated using Monte Carlo samples since it is written as an expectation. More precisely, given a set of sampled prompts $\theta$ from our posterior, we approximate the posterior predictive as $\frac{1}{m} \sum_{i=1}^m p( y_{\text{new}} \mid x_{\text{new}}, \theta^{(i)}), \qquad \theta^{(i)} \sim p(\theta \mid \mathcal{D})$. In practice, this amounts to generating a set of samples $ y^{(1)}, \ldots, y^{(m)} $ by running LLM passes with each prompt, i.e., by sampling $ y^{(i)} \sim p( y_{\text{new}} \mid x_{\text{new}}, \theta^{(i)}) $. This collection of samples can then be combined with various downstream uncertainty quantification methods.
>
> > A direct evaluation of “uncertainty” of the prediction distribution (Eq. 6) would be needed … calibration is more like a post-processing process.
>
> Given a set of samples $ y^{(1)}, \ldots, y^{(m)} $ from (Eq. 6), we do in fact evaluate how entropy is related to performance—in Appendix B.4, we have provided results using the popular Semantic Entropy (SE) [A] method applied on top of MHLP to perform abstention. Combining MHLP+SE outperforms all baselines.
>
> We also highlight that confidence/semantic confidence—our primary focus in the paper—are also uncertainty metrics themselves, and these are evaluated through calibration (Table 2) and abstention (Table 3). Actually, as you correctly point out, entropy and confidence are just two different post-processing steps on top of $ y^{(1)}, \ldots, y^{(m)} $ meant to summarize the uncertainty inherent in the samples.
>
> [A] Kuhn et al. “Semantic uncertainty: Linguistic invariances for uncertainty estimation in natural language generation” ICLR 2023.

---

> > ### Author Response · Authors · 2025-11-22
> >
> > > Is there any warm-up steps for prompt sampling?
> >
> > Yes, we do perform warm-up (also commonly referred to as burn-in) before prompt sampling. Please see Table 4 in Appendix B.2 for all hyperparameters around the burn-in steps.
> >
> > >  $\mathcal{D}$ is a single sample or an entire dataset?
> >
> > In full generality, $\mathcal{D}$ is the full training dataset. However, training sets are often large enough to make running a full pass over it per MCMC step intractable. Instead, we use mini-batching, which we explain in detail in Appendix A.1.

---

> > > ### Comment · Reviewer_j29K · 2025-11-22
> > >
> > > Thank you for your thorough response.\
> > > It really helps me understand the manuscript.\
> > > Since most of my concerns are alleviated with the authors' response, I raised my score from 4 to 6.\
> > > I hope our discussion would be beneficial to strengthen your work.

---

### Official Review · Reviewer_dut8 · 2025-10-29

**Soundness:** 3
**Presentation:** 4
**Contribution:** 3
**Rating:** 6
**Confidence:** 3

**Summary:**

This paper proposes a method called Textual Bayes, which treats a set of prompts as textual parameters $\theta$ in a Bayesian modeling $p(y|x, \theta)$. Specifically, $\theta$ contains a set of prompts, and the paper proposes an algorithm to optimize this set of prompts in order to maximize $p(D | \theta)$, where D is the data, so that we view the optimized set of prompts $\theta$ as faithful to the original prior (e.g., human-defined constraints), and best for model's performances on data $D$. The authors claim that using these prompts, we can better estimate the model's internal confidence by computing the expected probability over all prompts.

The core of this optimization algorithm is called Metropolis–Hastings through LLM Proposals (MHLP), which uses an LLM-powered optimization step (TextGrad) and the Metropolis–Hastings algorithm, which is a novelty of the paper. The paper also introduced semantic ECE, which can be used to calibrate free-form generations of LLMs. Experiments showed that the proposed Textual Bayes method led to better expected calibration error and better task performance across several datasets (AIME/SimpleQA/QASPER), and more reliable factual claims.

**Strengths:**

The overall setup offers a novel perspective on existing discussions in prompt selection, multi-step reasoning (e.g., self-reflection and TextGrad), and calibration, by treating textual prompts as Bayesian parameters. The authors provided a working recipe for this setup and demonstrated relatively promising results on calibration and factual correctness experiments. The paper is well organized, with a background section that helped me understand the overall setup in a short time. This paper may inspire future works to think more about abstracting LLM components into probabilistic frameworks.

**Weaknesses:**

1. The experiments are a little weak for the paper to be well-perceived as an uncertainty quantification work. The experiments are on three QA datasets with small sample counts. Some gains are modest and may not be statistically decisive (especially considering the variances). The baselines are a little weak, where the authors should consider prompt‑ensemble and decoding‑ensemble baselines (e.g., self‑consistency with diverse system messages/instruction styles, retrieval‑augmented ensembles, or perturbation and entropy-based methods). To showcase the effectiveness of the proposed prompt optimization, the authors should consider demonstrating examples and a human study, as well as comparing against simple baselines (e.g., having an LLM to "decompose" the original prompt into a diverse set of candidate system prompts that consider multiple angles).

2. The authors should cite other papers that similarly abstract textual factors into Bayesian parameters in LLM settings for training, inference, and uncertainty quantification that share high-level motivations. Example: BIRD: A TRUSTWORTHY BAYESIAN INFERENCE
FRAMEWORK FOR LARGE LANGUAGE MODELS

**Questions:**

See weaknessness.

---

> ### Author Response · Authors · 2025-11-22
>
> Thank you for your effort in providing this review. We are very glad to see you found our work “novel” with “promising results” and that the writing was “well organized”. We discuss both the weaknesses you mentioned below.
>
> > Some gains are modest and may not be statistically decisive
>
> It is true some gains are modest, but they hold across tasks and datasets. Moreover, all results include standard errors obtained over 10 independent runs; in Table 1-3 we bold the best performing methods within error bars to account for statistical deviations and find that our method still comes out clearly on top in most cases.
>
> > The baselines are a little weak, where the authors should consider prompt‑ensemble and decoding‑ensemble baselines (e.g., self‑consistency with diverse system messages/instruction styles, retrieval‑augmented ensembles, or perturbation and entropy-based methods).
>
> Thank you for also suggesting additional baselines, but we highlight that in fact our work does already cover most of these categories:
> - Paraphrasing and System-Message proposed by Gao et al. [A], for which we show results in Table 1-3 (see Appendix B.2.1 for more details), are self-consistency methods based on perturbation and diverse system messages, respectively. MHLP performs favourably compared to these baselines.
> - Entropy-based methods such as Semantic Entropy (SE) [B] are orthogonal to our method and can thus be applied on top of MHLP; we provide a discussion on this topic in L315-317 and Appendix B.4, and provide results using MHLP+SE in Table 11 showing that our method outperforms all baselines in this setting.
> - Retrieval-augmented ensembles are somewhat outside the scope of these question-answering tasks, but applying our Bayesian point of view with a RAG context would be an interesting subject for future work.
>
> [A] Gao et al. “SPUQ: Perturbation-based uncertainty quantification for large language models” EACL 2024
>
> [B] Kuhn et al. ”Semantic uncertainty: Linguistic invariances for uncertainty estimation in natural language generation” ICLR 2023.
>
> > The authors should cite other papers that similarly abstract textual factors into Bayesian parameters in LLM settings for training, inference, and uncertainty quantification that share high-level motivations.
>
> Thank you for the suggested citation; we will discuss it in our revision along with the suggestions provided by other reviewers. BIRD [C] does not perform Bayesian inference over textual parameters as we do, but instead extracts textual factors from a specific query and builds a Bayesian network over those factors. We will add this example to Sec. 5.
>
> [C] Feng et al. “BIRD: A Trustworthy Bayesian Inference Framework for Large Language Models” ICLR 2025

---

### Official Review · Reviewer_D3Rb · 2025-10-31

**Soundness:** 3
**Presentation:** 3
**Contribution:** 2
**Rating:** 6
**Confidence:** 4

**Summary:**

This paper aims at uncertainty quantification in LLM-based systems. Specifically, The authors interpret prompts as Bayesian textual parameters in a statistical model and perform Bayesian inference over these prompts with introduction of  Metropolis-Hastings through LLM Proposals (MHLP). These prompts are then used to calculate uncertainty. They demonstrate improved predictive accuracy and uncertainty quantification (UQ) on question-answering tasks and also in a conformal prediction setting.

**Strengths:**

The paper proposes a novel approach to quantify the uncertainty associated with prompts through Bayesian inference.

The narratives are generally clear and easy to follow.

**Weaknesses:**

The empirical improvements in Tables 1–3 are relatively small compared to the substantial computational overhead reported in Appendix Table 4. Since the evaluation is also conducted on very small datasets, this further weakens the strength of the empirical evidence.

I have concerns regarding the final sampled prompts. For example, I do not understand why the proposed prompt in step 20 is rejected. Another baseline that should be tested is just paraphrasing the prompt while keeping the question the same. The authors should also providing a full set of sampled prompts for at least one example which would help clarify this issue.

The justification regarding semantic entropy (lines 314–320) is not convincing. If multiple answers are semantically equivalent, they should be treated as a single answer. Similarly, in Tables 1 and 3 the evaluation should incorporate semantic clustering so that semantically equivalent answers are not counted as distinct ones.

**Questions:**

1. How exactly do the authors use suggestions from LLMs to propose values of θ that implement the guidelines? (Line 254) Could the authors show the actual prompt?

2. How are the textual constraints on prompts encoded in the prior, and how is it determined what constraints a valid prompt should satisfy? (Line 232)

3. In line 817, what does “substitute the final LLM call in our LLM-based system with an open-source model” specifically mean?

4. Missing citations: https://aclanthology.org/2024.naacl-long.390.pdf, https://arxiv.org/pdf/2412.09572

---

> ### Author Response · Authors · 2025-11-22
>
> Thank you for your time spent reviewing our paper! We are glad to see that you found our approach novel and the paper easy to follow. We will respond to your comments below.
>
> > Empirical improvements … are small compared to the substantial overhead (**W1**)
>
> Note that the additional overhead incurred by MHLP to sample prompts for a given task is only a one-time fixed initial “training” cost meaning that the extra LLM calls incurred by MHLP during MCMC sampling effectively become amortized at test-time.
>
> Moreover, while our empirical improvements are modest, they are consistent across datasets and UQ tasks, highlighting the overall promise of this research direction.
>
>
> >  I do not understand why the proposed prompt in step 20 is rejected. (**W2**)
>
> We agree this is a point worth discussing - as with any MCMC procedure, MHLP is a sampling algorithm that is stochastic such that the acceptance/rejection of prompts at each step is itself random; not all “good” proposed prompts will be accepted. Additionally, the effect of specific input perturbations (e.g., prompt rewording) on downstream performance is not always obvious with neural networks.
>
> > Another baseline that should be tested is just paraphrasing the prompt while keeping the question the same. (**W3**)
>
> We agree this baseline is sensible—in fact, System-Message proposed by Gao et al. [A] for which we show results in Table 1-3 (see Appendix B.2.1 for more details) is quite similar in spirit to your idea as a way of adding diversity to prompts. MHLP performs favourably compared to this baseline.
>
> [A] Gao et al. “SPUQ: Perturbation-based uncertainty quantification for large language models” EACL 2024
>
> > The justification regarding semantic entropy (lines 314–320) is not convincing. (**W4**)
>
> Admittedly, our phrasing here can be improved, and we will do so in our revision. To copy from our answer to Reviewer **jbfF**, the point we mean to make is that in our work (and many others), uncertainty is quantified through a 2-step pipeline:
> 1. Generate a diverse set of answers $ y^{(1)}, \ldots, y^{(m)} $ (by sampling multiple times from an LLM or using an ensemble of prompts as in MHLP and our System-Message baseline).
> 2. Summarize these answers into an uncertainty measure, such as semantic confidence or semantic entropy.
>
> Because ours is a technique for step 1, it is equally compatible with *any* step 2 method and hence they are orthogonal. Step 1 methods cannot be directly compared to step 2 methods, and so it does not make sense to compare MHLP and semantic entropy (whose standard implementation generates the answers in step 1 in an i.i.d. manner).
>
> However, MHLP and semantic entropy can be combined; this was shown in a small experiment in Appendix B.4, where MHLP with semantic entropy outperforms baselines by a wide margin in abstention.
>
> > [Semantically equivalent answers] should be treated as a single answer … the evaluation should incorporate semantic clustering… (**W5**)
>
> Indeed, we do use semantic clustering for evaluations of free-form datasets. As described in our work, (1) accuracy is estimated semantically using an LLM judge (L331), (2) expected calibration error is estimated using semantic confidence (semantic ECE L344 - L356), and (3) abstention ROC AUC is also estimated using the same semantic confidence scores (L338).
>
> > How exactly do the authors use suggestions from LLMs to propose values of θ that implement the guidelines? (Line 254) Could the authors show the actual prompt? (**Q1**)
>
> We implemented the UPDATE step, which proposes a new value of $\theta’$, using TextGrad (L270). This is described in more details in App A.2 where we also show an example prompt used by TextGrad:
> “You will be given a question related to scientific research papers and an answer attempted by a
>   language model. Evaluate the attempted answer. Be smart, logical, and very critical.
>   Do not solve the question. Just provide concise feedback”
> This example is for the QASPER dataset. Since each dataset represents a different task, the TextGrad prompt is created by hand for each dataset to describe the task’s objective. Each such prompt we used can be found in the config files of our supplementary material.
>
> > How are the textual constraints on prompts encoded in the prior, and how is it determined what constraints a valid prompt should satisfy? (Line 232) (**Q2**)
>
> The textual constraints are encoded as shown by using the output distribution of an LLM with a specific system prompt, and they are meant to encode one’s prior beliefs about the parameters of interest. Hence, a human would define these constraints **a priori** by expressing in natural language what they would expect a good prompt to look like. The constraints we use in practice are available in the config files in our supplementary material: they were fixed to be “The system prompt must be concise and generic and works with any question.” for all experiments.

---

> > ### Author Response · Authors · 2025-11-22
> >
> > > In line 817, what does “substitute the final LLM call in our LLM-based system with an open-source model” specifically mean? (**Q3**)
> >
> > To run our method MHLP and compute the likelihood we require access to log-probabilities on the output of the LLM-based system (LBS) (see L221). Hence, if the final call in the LBS is a blackbox, closed-source LLM, we use an open-source LLM as a surrogate model to compute the likelihood term in the acceptance ratio of MHLP, but not for running the LBS on the downstream task once prompts are sampled. This point from L817 was clarified more on L844.
> >
> > > Missing citations (**Q4**)
> >
> > Thank you for pointing out additional works which can provide more context on different approaches to uncertainty quantification. We will cite and discuss them in Section 5 in the revision. The first [B] is focused on detecting hallucinations and uses multiple generations to judge uncertainty on a given output. The system prompt is modified by hand to encourage more diversity in the additional generations. The second [C] also uses self-consistency evaluation for hallucination detection, but focuses on the uncertainty of an LLM’s parametric knowledge. Multi-agent interactions are used to generate diverse instantiations of a query, but the system prompt is unchanged. In contrast to both of these works, our goal is to use Bayesian ideas to quantify uncertainty over textual parameters or downstream predictions. We specify a prior over prompts to sample them, rather than editing by hand, and do not modify the query.
> >
> >
> > [B] Zhao et al. “Knowing What LLMs DO NOT Know: A Simple Yet Effective Self-Detection Method” NAACL 2024
> >
> > [C] Feng et al. “Rethinking LLM Uncertainty: A Multi-Agent Approach to Estimating Black-Box Model Uncertainty” EMNLP Findings 2025

---

### Official Review · Reviewer_jbfF · 2025-10-31

**Soundness:** 3
**Presentation:** 2
**Contribution:** 3
**Rating:** 2
**Confidence:** 3

**Summary:**

The authors propose Textual Bayes to quantify uncertainty in black-box LLM systems by treating prompts as Bayesian textual parameters. It introduces a new MCMC algorithm, MHLP, which uses the LLM to generate prompt proposals improving uncertainty quantification and predictive accuracy.

**Strengths:**

- The central idea of treating the prompt ($\theta$) as a textual parameter in a formal Bayesian model ($p(\theta|\mathcal{D})$) is interesting. It provides a principled, statistical approach to modeling the brittleness of LLM systems, namely their sensitivity to prompt engineering.
- The proposed MHLP algorithm is theoretically well motivated. It repurposes prompt optimization techniques as MCMC proposal distributions for UQ even when using closed-source, black-box APIs.

**Weaknesses:**

**Motivation:** The claim "we are the first both to quantify the uncertainty associated with prompts in LLM-based systems" (line 79) is not correct! There exist multiple works that have already considered this, such as Abbasi-Yadkori et al. [1], Tonolini el al. [2], or Zhang et al. [3].

**Method**: The authors introduce a sophisticated and complex theoretical methodology. However, the proposed practical implementation relies on crude approximations that compromise the Bayesian guarantees and that questions whether a much simpler, cheaper heuristic would perform just as well:
1. Correction Factor: The required Hastings Correction is ignored (set to 1) because black-box LLMs provide no probabilities. This transforms the method from a theoretically sound sampler into an ad-hoc Metropolis algorithm.
2. Acceptance Metric: The core quality measure (the likelihood term) is calculated by a different surrogate model. Moreover, to make the method computationally feasible, it is approximated over a singe sample (batch size set to 1 as stated in in line 283).
The complex MCMC process ultimately reduces to accepting a new prompt if a single sample yields a higher log-likelihood score under a surrogate model.

**Evaluation**:
1. The uncertainty being quantified is primarily over prompt choice rather than epistemic or aleatoric uncertainty in the traditional sense. This should be made more clear and also reflected in the title as it might be misleading.
2. The authors claim that "all methods use the same number $m$ of LBS calls during inference to ensure a fair comparison from a computational perspective" (line 312). This is technically true but highly misleading: the methods use the same inference budget (10 calls per test instance) but they do *not* use the same total budget. MHLP gets 180-300 extra calls that CoT doesn't get.


---
[1] Y. Abbasi-Yadkori, Ilja Kuzborskij, A. György, and C. Szepesvari. To believe or not to believe your LLM: iterative prompting for estimating epistemic uncertainty. NeurIPS, 2024.

[2] F. Tonolini, N. Aletras, J. Massiah, and G. Kazai. Bayesian Prompt Ensembles: Model Uncertainty Estimation for Black-Box Large Language Models. ACL, 2024.

[3] Z. Zhang, A. Verma, F. Doshi-Velez, B. Low. Understanding the Relationship between Prompts and Response Uncertainty in Large Language Models. arXiv preprint arXiv:2407.14845, 2024.

**Questions:**

- The argument for not computing an uncertainty score (such as SE), namely that the "method is a means of generating a diverse answer set" (line 314), is not clear. Why does MHLP lead to diverse answers? And why can't an uncertainty score be computed that is then compared to SE and other UQ methods?
- Re. Evaluation point 2: What if CoT gets the same overall budget of MHLP? Would MHLP with matched compute still underperform this baseline?

---

> ### Author Response · Authors · 2025-11-22
>
> Thank you for your review. We are glad you found our work interesting, principled, and theoretically well-motivated and that you highlighted the ability of our method in quantifying UQ in closed-source, blackbox LLMs.
>
> Here, we answer your concerns one by one. We believe these answers fully address all or at least most of your concerns; if you agree, we hope you consider raising your score. Otherwise, please let us know, and we will endeavor to resolve anything further.
>
> **Motivation**
> > The claim "we are the first both to quantify the uncertainty associated with prompts in LLM-based systems" (line 79) is not correct!
>
> Thank you for sharing these works, all of which are relevant. We will include them in our updated manuscript.
>
> In light of these works, especially [2], we agree that this claim needs to be toned down. In particular, [2] performs Bayesian inference over a set of weights associated with a small set of pre-selected prompts.
>
> We will thus tone it down to the following, which still holds:
>
> “To the best of our knowledge, we are the first to perform Bayesian inference over the space of *free-form prompts* in LLM-based systems.”
>
> **Method**
>
> > However, the proposed practical implementation relies on crude approximations that compromise the Bayesian guarantees …to make the method computationally feasible, it is approximated over a single sample …
>
> While it is true that our method does use approximations, this is also generally true of Bayesian inference and MCMC methods in deep learning settings, which are unfailingly complex and high-dimensional. Bayesian inference is almost never performed perfectly accurately—e.g., MCMC methods are never run for infinitely many steps and the variational family in variational inference seldom includes the true posterior. We make certain approximations throughout, but necessarily so. Importantly, despite these approximations, we still outperform all baselines.
>
> > The required Hastings Correction is ignored
>
> This is actually not the case—we do estimate the Hastings correction $\frac{q(\theta^{(t-1)} \mid \theta')}{q(\theta' \mid \theta^{(t-1)})}$. This is made possible by using open-source LLMs (e.g., LLaMa-3.1-NemoTron-70B-Instruct-HF) in the proposal $q(\cdot \mid \cdot)$. This is described in Appendix A.2, but we apologize for neglecting to mention it in the main text—we will point to this section in the main text of our revision.
>
> > The core quality measure (the likelihood term) is calculated by a different surrogate model.
>
> This is true in our experiments, and is made necessary because we focus on closed source, blackbox LLMs. However, this is not necessarily a requirement of our method when the model’s log-probabilities are available.
>
> To demonstrate this feature, we have run some additional experiments where we instead used the open-source Llama-3.1-Nemotron-70B-Instruct-HF outright as the LLM (instead of GPT), thus removing the need for a surrogate model altogether. We show accuracy and ECE results on the AIME 2024 dataset and compare with our main baselines also using the same open-source LLM. Similar to our main findings using GPT with the surrogate model, we see that MHLP continues to outperform baselines when no surrogate model is used as well, demonstrating the generality of our approach.
>
> Methods with no surrogate model using Llama-3.1-Nemotron-70B-Instruct-HF:
>
> | Method  | ACC ↑  | ECE ↓  |
> |---|---|---|
> | CoT  | 12.8 ± 0.7 | 49.3 ± 1.1 |
> | TextGrad   | 20.6 ± 1.0 | 26.1 ± 1.6 |
> | MHLP | **22.9 ± 1.1** | **22.7 ± 1.3** |
>
>
> **Evaluation**
>
> > The uncertainty being quantified is primarily over prompt choice rather than epistemic or aleatoric uncertainty in the traditional sense. This should be made more clear and also reflected in the title as it might be misleading.
>
> We agree with your point, and indeed the traditional epistemic/aleatoric uncertainty paradigm does not fit our method well. Reviewer **j29K** also pointed this out.
>
> To address this, we will change the title to “Textual Bayes: Quantifying *Prompt* Uncertainty in LLM-Based Systems”. We will also add language differentiating uncertainty in our context (in prompts, over LLM-based systems) from the usual context (over, e.g., model weights or other choices).
>
> (Response continued below)

---

> > ### Author Response · Authors · 2025-11-22
> >
> > > What if CoT gets the same overall budget of MHLP? Would MHLP with matched compute still outperform this baseline? (**W2 + Q2**)
> >
> > While we have made our best effort to make all comparisons fair, there is no straightforward way to equalize the overall budget between CoT and MHLP.
> > - On the one hand, CoT is a technique that is applied to each LLM call at test-time.
> > - On the other hand, the additional cost incurred by MHLP to sample prompts is only a one-time fixed initial “training” cost per task meaning that the extra LLM calls incurred by MHLP during MCMC sampling effectively become amortized at test-time.
> >
> > Simply adding more LLM calls to the inference budget of CoT to compensate for this would not equalize their overall budgets; this would in fact make the total budget of CoT far greater.
> >
> > However, we can still investigate this question in spirit through our *TextGrad* baseline, which is a CoT prompt + some extra initial compute used to refine this prompt. We can increase the number of TextGrad steps to increase this initial one-time compute usage, thus representing a way of adding more compute on top of CoT. We find that increasing compute does not necessarily improve performance for TextGrad + CoT, and that MHLP uses this compute budget more effectively.
> >
> > | Method  | Steps | Accuracy | ECE  | Abstention ROC |
> > |--|--|--|--|--|
> > |  TextGrad + CoT  | 20  | 58.8 ± 1.0  | 21.6 ± 1.2  | 67.4 ± 0.9 |
> > |  TextGrad + CoT  | 40  | 56.9 ± 1.8  | 26.0 ± 2.0  | 68.6 ± 1.2 |
> > |  TextGrad + CoT  | 60  | 55.4 ± 1.8  | 26.3 ± 0.6  | **69.2 ± 3.6**  |
> > |  TextGrad + CoT  | 80  | 52.3 ± 1.5  | 25.5 ± 1.3  | **71.5 ± 5.4**  |
> > | MHLP  | 20  | **60.9 ± 1.0** | **17.7 ± 1.1** | **71.7 ± 0.9**  |
> >
> > Note that, due to time constraints, the middle 3 rows are averaged over 3 runs, while the first and last (taken from the paper) are computed over 10. The trend is nevertheless quite clear, with TextGrad + CoT accuracy actually decreasing with additional compute.
> >
> >
> > > The argument for not computing an uncertainty score .. is not clear. Why does MHLP lead to diverse answers? And why can't an uncertainty score be computed that is then compared to SE and other UQ methods? (**Q1**)
> >
> > Thank you for raising this point, which Reviewer D3Rb also touched on. We will work the following clarifications into our revision.
> >
> > The point we mean to make is that in our work (and many others), uncertainty is quantified through a 2-step pipeline:
> > 1. Generate a diverse set of answers $y^{(1)}, \ldots, y^{(m)}$ (by sampling multiple times from an LLM or using an ensemble of prompts as in MHLP and our System-Message baseline).
> > 2. Summarize these answers into an uncertainty measure, such as semantic confidence or semantic entropy.
> >
> > Because ours is a technique for step 1, it is equally compatible with *any* step 2 method and hence they are orthogonal. Step 1 methods cannot be directly compared to step 2 methods, and so it does not make sense to compare MHLP and semantic entropy (whose standard implementation generates the answers in step 1 in an i.i.d. manner).
> >
> > However, MHLP and semantic entropy can be combined; this was shown in a small experiment in Appendix B.4, where MHLP with semantic entropy outperforms baselines by a wide margin in abstention.

---

### Official Review · Reviewer_wYh1 · 2025-11-01

**Soundness:** 3
**Presentation:** 3
**Contribution:** 3
**Rating:** 6
**Confidence:** 4

**Summary:**

This paper presents a Textual Bayes, a Bayesian framework for uncertainty quantification in LLM-based systems. The key idea of this work is how to treat prompts for LLMs as training textual parameters and perform Bayesian inference over them. For this, authors introduce a novel algorithm Metropolis–Hastings through LLM Proposals (MHLP), which leverages LLM-driven prompt optimization to generate proposal prompts and sample from a posterior distribution. It allows to estimate uncertainty over both prompts and outputs.

Experimental results on QA and factuality benchmarks show that proposed framework, Textual Bayes, improves the following metrics: predictive accuracy, calibration (via lower Expected Calibration Error and a new semantic ECE metric). This method also improves factual reliability in conformal prediction settings by reducing the number of removed claims. Overall, the authors introduce novel integration of Bayesian reasoning with black-box LLM-based systems, enhancing uncertainty quantification in generative AI.

**Strengths:**

1. Recasts prompts as Bayesian textual parameters in \( p(y \mid x, \theta) \) and performs inference over prompts \(\theta\).
2. Introduces MHLP, an MCMC scheme with LLM-driven proposals – both novel and well-motivated.
3. Clear prior construction via free-form textual constraints.
4. Proposal mechanism targets high-posterior prompts by respecting priors and improving data likelihood \( p(D \mid \theta) \).
5. Demonstrates consistent gains in accuracy and calibration across QA benchmarks.

**Weaknesses:**

1. High computational cost due to MCMC with large LLMs. It is worth considering lighter models.
2. The paper explicitly uses a cold/hot posterior and surrogate/open-source likelihoods for closed-source models, but doesn’t show sensitivity to these choices, which could bias the posterior and calibration.
3. Evaluation limited to single-prompt systems; no empirical tests with multiple textual parameters \((\theta_1, \ldots, \theta_k)\).
4. Limited study of prior quality and initialization (limited analysis of sensitivity to priors and initialization)

**Questions:**

1.	The paper mentions applying a “cold/hot posterior” using a likelihood temperature \( \tau \) (or the equivalent scaling factor \( \beta \)), but does not explain the choice or its impact. Could you clarify what specific temperature values were used, how they were selected or tuned, and whether model accuracy or calibration metrics (e.g., ECE) are sensitive to different temperature settings?
2.	Could you report standard diagnostics for the MHLP sampling process, such as acceptance rates, effective sample sizes, or any convergence checks (e.g., monitoring of likelihood or posterior stability)? Additionally, did you test whether using longer Markov chains (more MHLP steps) materially changes the results in terms of accuracy or calibration?
3.	How do outcomes change with weaker/stronger or mis-specified textual priors \\(p(\theta)\\)? Did you run any ablations that vary the prior constraints versus the data likelihood \\( p(D \mid \theta) \\) to assess sensitivity?
4.	Can MHLP handle multiple textual parameters \\( \theta = (\theta_1, \ldots, \theta_k) \\) within a single system pass, and have you tested this capability in agentic or multi-stage LLM pipelines?
5.	The paper references the use of a “cold/hot posterior” and defines a tempered likelihood scaling factor \\( \beta := \frac{n}{\tau b} \\), but the description appears inconsistent – Section 3 mentions a cold posterior, while Appendix A.1 describes an effective “hot” behavior when \\( \beta < n \\).  Please clarify whether the posterior is cold, hot, or generally tempered; explicitly report the values of \\( \tau \\) (or \\( \beta \\)) used in experiments and how they were selected.  In addition, please interpret these quantities relative to the mini-batch size \\( b \\), not to the total dataset size \\( n \\), to make the scaling consistent and reproducible.

---

> ### Author Response · Authors · 2025-11-22
>
> Thank you for spending time on this thoughtful review. We are glad you found the work novel and well-motivated! We answer your questions and address the weaknesses you raised below.
>
> > High computational cost due to MCMC with large LLMs. It is worth considering lighter models. (**W1**)
>
> You are correct that MCMC is more costly when used with large LLMs and this is a fundamental constraint in prompt optimization-type methods. For completeness’ sake we point out that
> - **This is a one-time fixed “training” cost,** after which MCMC prompt samples are obtained that can be reused ad infinitum. The initial MCMC cost is thus effectively amortized at test-time.
> - **We do consider lighter models**: our results on the QASPER dataset did use a smaller and lighter model: GPT-4o mini.
> - Our use of large LLMs was intentional to highlight a major benefit of our method: compatibility with blackbox LLMs, for which there are very limited ways to quantify uncertainty. We aimed to showcase our method with these models that tend to be large (e.g., GPT-4o).
>
> > Clarity about tempered posterior (**W2 + Q1 + Q5**)
>
> Thank you for pointing out that we could have been more clear around cold/hot posteriors. We will incorporate these resolutions into our upcoming revision. We generally refer to a “tempered” posterior when it is taken to the power $1/\tau$ (see L785 in App A.1 for full details). Often in the literature the temperature parameter is “cold” ($0\leq \tau \leq 1$), so we used this terminology when referencing past work to guide the reader’s search for more background (e.g., on L272 in Sec. 3). In practice, we treated the temperature as a hyperparameter and tuned it for performance (L880), and we did report the final values used in Table 4. Larger values of $\tau$ were more performant in our experience, hence why we mentioned a “hot” posterior in L795. In experiments we used a mini-batch size of 1 (L783) so it is appropriate to interpret these quantities relative to the total dataset size.
>
> > Concerns about sensitivity to choice of surrogate/open source likelihoods for closed-source models (**W2**)
>
> To answer your question about the comparison between surrogate and non-surrogate models, we have run some additional experiments where we instead used the open source Llama-3.1-Nemotron-70B-Instruct-HF outright as the LLM (instead of GPT), thus removing the need for a surrogate model altogether. We show accuracy and ECE results on the AIME 2024 dataset and compare with our main baselines also using the same open-source LLM. Similar to our main findings using GPT with the surrogate model, we see that MHLP performs favourably when no surrogate model is used as well, demonstrating the generality of our approach.
>
> | Method  | ACC ↑  | ECE ↓  |
> |---|---|---|
> | CoT  | 12.8 ± 0.7 | 49.3 ± 1.1 |
> | TextGrad   | 20.6 ± 1.0 | 26.1 ± 1.6 |
> | MHLP | **22.9 ± 1.1** | **22.7 ± 1.3** |
>
> > Can MHLP handle multiple textual parameters $\theta = (\theta_1, \ldots, \theta_k)$ (**W3 + Q4**)?
>
> Thank you for raising this point. Indeed, MHLP can handle multiple textual parameters in an arbitrarily complex LLM-based system composed of several LLM calls. We have run an additional experiment where we consider Self-Refine [A], a self-reflecting LLM-based system consisting of a three stage pipeline: generation, feedback, and refinement, each stage with its own system prompt. On the QASPER dataset we apply TextGrad to optimize all three prompts, and compare it to MHLP with three textual parameters in a single LLM-based system. MHLP achieves the best overall performance across all metrics.
>
> | Method  | ACC ↑  | ECE ↓  | ROC AUC ↑  |
> |---|---|---|---|
> | Self-Refine | **58.5 ± 1.0** | 25.9 ± 0.7 | **72.9 ± 1.0** |
> | TextGrad | 56.5 ± 0.9 | **20.6 ± 1.0** | 68.1 ± 1.2 |
> | MHLP | **59.2 ± 0.9** | **22.1 ± 1.1** | **74.2 ± 1.2** |
>
> [A] Madaan et al. “Self-Refine: Iterative Refinement with Self-Feedback” NeurIPS 2023
>
> > Limited study of prior quality and initialization (**W4 + Q3**)
>
> We refrained from tuning the prior based on experimental results, because this is not in the spirit of a Bayesian approach. We used the same prior distribution (the output of an LLM given textual constraints) specified in Eq. 7 across all experiments for consistency and to show that MHLP works “out-of-the-box” without requiring the practitioner to spend effort crafting this distribution. We also initialized both TextGrad and MHLP to the same CoT prompt as our CoT baseline (L304 and L308-310) to ensure a more direct comparison.

---

> > ### Author Response · Authors · 2025-11-22
> >
> > > Could you report standard diagnostics for the MHLP sampling process? (**Q2**)
> >
> > Thank you for raising this—though we omitted mention of it in the main text, we did report some diagnostics on the MHLP sampling process in Appendix B.3. We will make this clearer in our manuscript revision.
> >
> > Figure 3 in particular shows the acceptance rate over the number of MCMC steps as you requested. Interestingly, the acceptance rate settles down to the theoretically optimal value given in previous studies (see L1105), which gives us more confidence that MHLP sampling can converge in practice. In informal tests we found that longer Markov chains (more MHLP steps) did not improve performance, but running comprehensive ablations on this parameter would be too expensive for us to perform at this time given API costs.

---

### Author Response · Authors · 2025-12-03
**Note for AC**

Dear AC,

Thank you for taking on our paper! We have uploaded our latest manuscript revision incorporating reviewer suggestions, and would like to point out that:
1. Reviewer **j29K** **raised their score from 4 -> 6** on Nov. 22nd before the OpenReview leak was widely known, as they mention in their response ("Since most of my concerns are alleviated with the authors' response, I raised my score from 4 to 6.”). We thank reviewer **j29k** for engaging on this and now for recommending the acceptance of our paper!

2. Although reviewer **jbfF**—who gave us our other reject score—did not get a chance to respond before reviewer comments were paused, we do believe we have addressed all of their concerns to a satisfactory degree. Their concerns and our responses can be summarized as follows:

**Writing**
 -  We over-claim in a sentence in our intro:
     - In response, we have adjusted this claim’s verbiage and cited the relevant sources.
 - The Hasting correction term in MCMC is ignored by our algorithm:
     - This is actually not true—we do include the Hastings correction, and we have clarified this in our revised manuscript.
- “Quantifying uncertainty” can be misleading in that our method mainly focuses on uncertainty over prompts rather than traditional epistemic/aleatoric uncertainty:
   - We have reflected this in a change to our paper’s title.
- Why is our method not directly comparable to semantic entropy (SE) type methods?:
   - We have clarified in our manuscript that SE-type methods are a different step of the uncertainty pipeline and are hence orthogonal; we also highlighted an existing experiment in our Appendix in which we combine our method with SE.

**Method**
- Approximations in our method compromise its Bayesian guarantees:
     - While this is true, we point out that all Bayesian deep learning methods rely on heavy approximations for tractability; we were forthcoming about this in our manuscript.
 - In particular, a surrogate model is required to compute the likelihood values for MCMC:
     - This is true *in our experiments* which intentionally focus on closed-source blackbox models without available likelihoods, but not in our method in full generality. We have provided a small experiment for the reviewer to show that our method continues to outperform baselines in a fully open-source context where likelihoods are available.
- Concerns about whether compute budgets have been equalized between our method and our chain-of-thought (CoT) baseline:
   - We point out that it is not technically possible to equalize compute budgets between our method and CoT. We provide an experiment attempting to approximate adding more budget to CoT and find that our method continues to outperform baselines.

---

### Meta-Review · Area_Chair_cyCz · 2026-01-06

**Summary:**

Most concerns are resolved by the rebuttal, especially it is very helpful that the reviewers managed to point out the previous inappropriate position of the work, and the authors agree to clarify and change their framing of the contribution. It is important to update the paper to reflect an appropriate comparison against previous Bayesian works in LLM UQ and differentiate with conventional uncertainty notions that focus on the model.

In addition, I suggest the authors include more discussion on the higher computational cost and expand their evaluation to more diverse tasks and more comprehensive metrics.

**Reviewer Concerns:**

The main concerns of the reviewers included: 1) inappropriate position of the work, against previous Bayesian works in LLM UQ and limited clarification and discussion against conventional uncertainty notions; 2) higher computational cost; 3) relation with uncertainty estimation method that focuses on semantic equivalence; 4) limited evaluation on small QA tasks.

I would say 1) and 3) are mostly resolved by the rebuttal, but 2) and 4) are still somewhat outstanding.

**Reviewer Scores:**

Reviewer j29K mentioned that they will raise the score from 4 to 6.

I predict reviewer wYh1 may increase the score to maybe 4, given the authors fully accepted the criticism that the previous framing of the contribution and the title is misleading. They will put their work in a more appropriate position in the landscape of the literature.

The other two reviewers may maintain their score.

---

### Decision · Program_Chairs · 2026-01-26

Accept (Poster)